# Casein kinase 1G2 suppresses necroptosis-promoted testis aging by inhibiting receptor-interacting kinase 3

Dianrong Li[1,2†], Youwei Ai[1,2†], Jia Guo[1,2], Baijun Dong[3], Lin Li[1,2], Gaihong Cai[1,2], She Chen[1,2], Dan Xu[1,2], Fengchao Wang[1,2], Xiaodong Wang[1,2*]

[1]National Institute of Biological Sciences, Beijing, China; [2]Tsinghua Institute of Multidisciplinary Biomedical Research, Tsinghua University, Beijing, China; [3]Department of Urology, Renji Hospital, School of Medicine, Shanghai Jiao Tong University, Shanghai, China

**Abstract** Casein kinases are a large family of intracellular serine/threonine kinases that control a variety of cellular signaling functions. Here we report that a member of casein kinase 1 family, casein kinase 1G2, CSNK1G2, binds and inhibits the activation of receptor-interacting kinase 3, RIPK3, thereby attenuating RIPK3-mediated necroptosis. The binding of CSNK1G2 to RIPK3 is triggered by auto-phosphorylation at serine 211/threonine 215 sites in its C-terminal domain. CSNK1G2-knockout mice showed significantly enhanced necroptosis response and premature aging of their testis, a phenotype that was rescued by either double knockout of the *Ripk3* gene or feeding the animal with a RIPK1 kinase inhibitor-containing diet. Moreover, CSNK1G2 is also co-expressed with RIPK3 in human testis, and the necroptosis activation marker phospho-MLKL was observed in the testis of old (>80) but not young men, indicating that the testis-aging program carried out by the RIPK3-mediated and CSNK1G2-attenuated necroptosis is evolutionarily conserved between mice and men.

**\*For correspondence:**
wangxiaodong@nibs.ac.cn

[†]These authors contributed equally to this work

**Competing interests:** The authors declare that no competing interests exist.

**Reviewing editor:** Jennifer Garrison,

## Introduction

RIPK3 is an intracellular serine/threonine kinase with a key role in necroptosis, a regulated form of necrotic cell death (*Christofferson and Yuan, 2010*; *Vandenabeele et al., 2010*; *Wallach et al., 2016*). During necroptosis, in response of TNF-family of cytokines, toll-like receptors (TLRs), and Z-RNAs, RIPK3 is activated either by a related kinase RIPK1, or adaptor proteins TRIF, or ZBP1/DAI, respectively (*Cho et al., 2009*; *Degterev et al., 2008*; *He et al., 2009*; *Zhang et al., 2009*; *Zhang et al., 2020*). Activated RIPK3 phosphorylates the mixed lineage kinase-domain-like protein, MLKL, to trigger its oligomerization and translocation from the cytosol to membranes including plasma membrane for their disruption (*Cai et al., 2014*; *Chen et al., 2014*; *Sun et al., 2012*; *Wang et al., 2014*). Necroptosis is actively suppressed by caspase-8-mediated cleavage of RIPK1 and RIPK3, whose upstream activation pathway is often shared between caspase-8 and RIPK1 kinase. Embryonic lethality in *Caspase-8* knockout mice is rescued by double knockout *Ripk3* or *Mlkl,* and cellular necroptosis induction by TNF-α or TLRs is dramatically enhanced if caspase-8 activity is suppressed (*Dannappel et al., 2014*; *Dillon et al., 2014*; *Günther et al., 2011*; *He et al., 2009*; *Kaiser et al., 2011*; *Newton et al., 2019*; *Oberst et al., 2011*; *Rickard et al., 2014*; *Takahashi et al., 2014*). RIPK3 was also reported to be negatively regulated by the phosphatase Ppm1b but the effect seems minor and the in vivo validation has yet to be obtained (*Chen et al., 2015*).

One of the important physiological function of necroptosis is to promote the aging of testis in mice (*Li et al., 2017*). The necroptosis activation marker, the phosphorylated MLKL has been

observed in spermatogonium stem cells and Sertoli cells in the seminiferous tubules of old (>18 months) but not young mouse testis. Knockout *Ripk3*, or *Mlkl*, or feeding mice with a RIPK1 kinase inhibitor-containing diet, blocks necroptosis from occurring in mouse testis, and allows mice to maintain the youthful morphological features and the function of male reproductive system to advanced ages, when age-matched wild-type mice had lost their reproductive function (*Li et al., 2017*).

Casein kinases are a large family of intracellular serine/threonine kinases that control a variety of cellular signaling functions that include the circadian clock, Wnt receptor activation, μ opioid receptor modulation, DNA repair, and hypoxia response (*Davidson et al., 2005*; *Elyada et al., 2011*; *Etchegaray et al., 2009*; *Goldberg et al., 2017*; *Pangou et al., 2016*). We found several of the casein kinase (CSNK) 1 family members, including CSNK1D1, CSNK1E, and CSNK1G2, bind to and inhibit RIPK3 kinase activity. Interestingly, CSNK1G2 expresses at the highest level in mouse testis, whose expression pattern overlaps with that of RIPK3. Knocking out *Csnk1g2* in mouse or multiple cell lines including cell lines derived from spermatocyte and Sertoli cells significantly enhanced necroptosis response and the *Csnk1g2* knockout mice showed premature aging of their testis. Our results demonstrate that CSNK1G2 is a major negative regulator of necroptosis.

## Results

### CSNK1G2 negatively regulates necroptosis by binding to RIPK3

In a course of investigating RIPK3-interacting proteins, we found that several members of the casein kinase 1 family were among the proteins co-precipitated with RIPK3 kinase (*Figure 1—figure supplement 1A*). The effect of CSNK1 members on RIPK3 kinase activity was then assessed by co-expressing each member with RIPK3 in human embryo kidney 293 T cells, and the RIPK3 kinase activity was measured by probing the serine 227 auto-phosphorylation of RIPK3, an event critical for RIPK3 to recruit its substrate MLKL (*Li et al., 2015*; *Sun et al., 2012*). Among the casein kinase family members, CSNK1D1, CSNK1G2, and CSNK1E suppressed serine 227 phosphorylation on RIPK3 (*Figure 1—figure supplement 1B*). In particular, CSNK1G2, but not its closest family members CSNK1G1 and CSNK1G3, showed the most potent suppression of RIPK3 kinase activity (*Figure 1—figure supplement 1C*). Two kinase-dead mutants, CSNK1G2(K75A) and CSNK1G2(D165N), failed to suppress RIPK3 kinase activity (*Figure 1A*), indicating that the kinase activity of CSNK1G2 is required for its function in suppressing RIPK3. Although CSNK1G2 was not among the casein kinases co-precipitated with RIPK3 in 293 T cells, likely due to lack of expression in this cell line, its ability to strongly inhibit RIPK3 kinase activity and its pattern of tissue expression (see below) prompted us to invest this isoform of casein kinase further. Consistent with its ability to inhibit RIPK3 activity when co-expressed in 293 T cells, knockout *Csnk1g2* in mouse embryonic fibroblasts, MEFs, significantly accelerated MEF necroptosis induced by the combination of T̲NF-α (T), a S̲mac mimetic (S), and a pan-caspase inhibitor Z̲-VAD-fmk (Z) (*Figure 1B*). The enhanced necroptosis was mitigated by reintroducing wild-type CSNK1G2 into the *Csnk1g2* knockout MEFs, but a similar level of K75A kinase-dead mutant did not restore the necroptosis inhibition activity (*Figure 1B*). In addition to TSZ, MEFs with their *Csnk1g2* knocked out also showed enhanced cell death when treated with death-inducing cytokine TRAIL plus a Smac mimetic and z-VAD (TRAIL/S/Z), or lipopolysaccharide (LPS) plus a Smac mimetic and z-VAD (LPS/S/Z) (*Figure 1—figure supplement 2A*).

To further explore the mechanism through which CSNK1G2 suppresses necroptosis, MEFs with their endogenous *Csnk1g2* knocked out or rescued with wild-type or kinase-dead CSNK1G2 cDNA were treated with necroptosis-inducing T/TRAIL/LPS+S+Z, and the necroptosis activation markers phospho-RIPK3 (at threonine 231 and serine 232, equivalent to serine 227 of human RIPK3) and phospho-MLKL (at serine 345, equivalent to serine 358 of human MLKL) were analyzed by western blotting. Knocking out *Csnk1g2* in MEFs resulted in higher levels of phospho-RIPK3 and phospho-MLKL (*Figure 1C*, lanes 5 and 6, and *Figure 1—figure supplement 2B*). Reintroducing wild-type, but not kinase-dead, mutant CSNK1G2, significantly decreased the levels of phospho-RIPK3 and phospho-MLKL when cells were treated with necroptosis inducers (*Figure 1C*, lanes 7 and 8).

Necroptosis induced by TSZ is initiated by the formation of necrosome, a protein complex containing both RIPK1 and RIPK3 (*Cho et al., 2009*; *He et al., 2009*; *Zhang et al., 2009*). MEFs with their *Csnk1g2* knocked out showed more RIPK1 kinase association with RIPK3 (*Figure 1D*, lanes 2

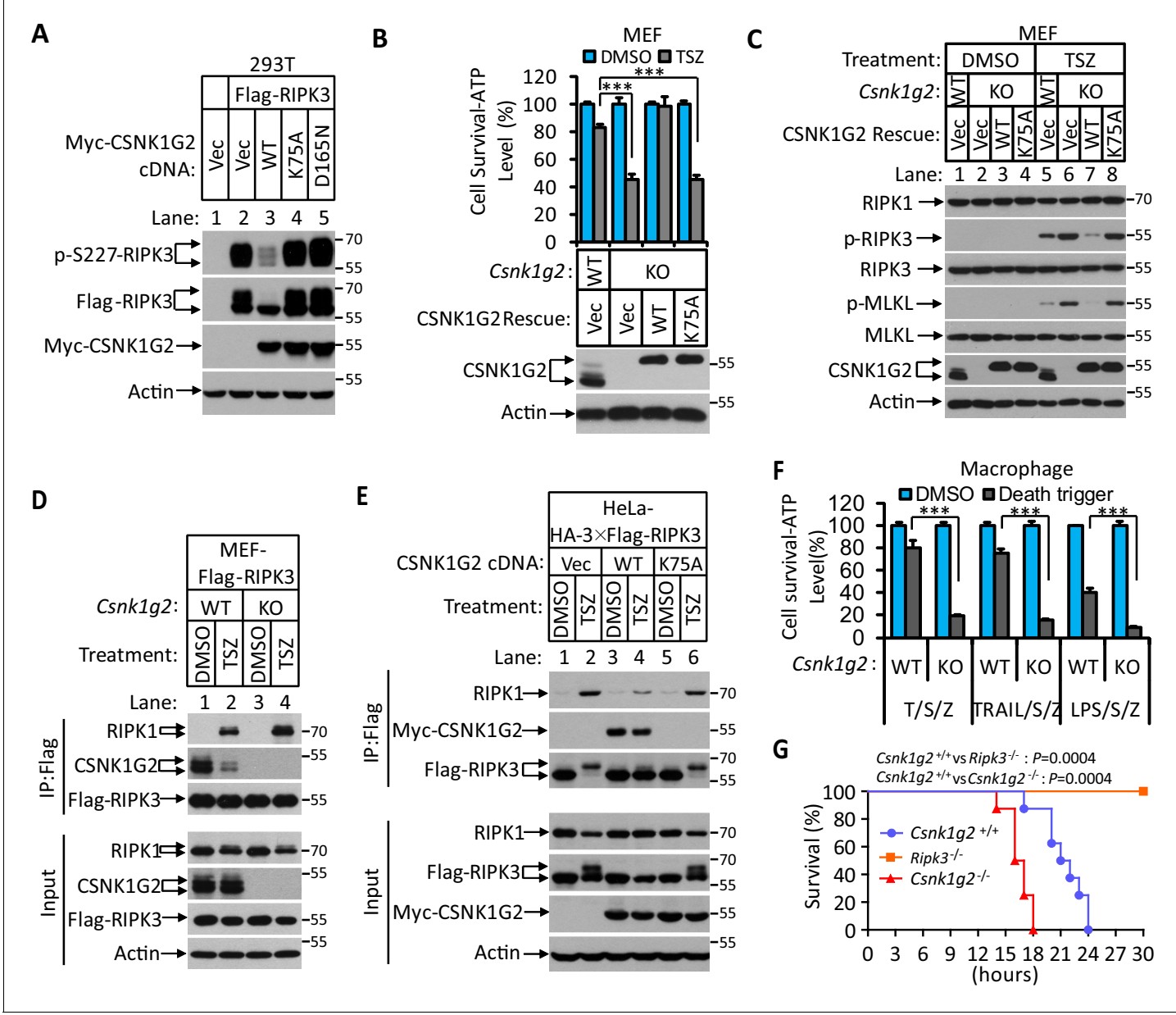

**Figure 1.** Knockout *Csnk1g2* accelerates necroptosis. (**A**) Western blotting analysis using antibodies against the indicated proteins. Cultured 293 T cells were transfected with Flag-tagged RIPK3 and the indicated versions of Myc-tagged CSNK1G2, including wild-type (WT) and two kinase-dead point mutants K75A and D165N for 20 hr. Cell extracts were then prepared and used for western blotting analysis. Vec, vector control. Numbers on the right indicate molecular weight markers (kDa). (**B**) Top: Cell viability as measured by Cell Titer-Glo. Cultured MEF with wild-type *Csnk1g2* gene (WT) or with their *Csnk1g2* gene knocked out (KO) followed by transfection with vector control (Vec) or indicated wild-type or a kinase-dead (K75A) mutant CSNK1G2 MEF. The cells were then treated with DMSO or TSZ as indicated for 12 hr before the intracellular ATP levels were measured by Cell Titer-Glo. T denotes 20 ng/ml TNF-α; S, denotes 100 nM Smac mimetic; Z denotes 20 μM Z-VAD-FMK. Data are mean ± SD of triplicate wells. ***p<0.001. p-values were determined by two-sided unpaired Student's *t*-tests. Bottom: Aliquots of these treated cells were used for western blotting analysis using an antibody against CSNK1G2. (**C**) Western blotting of necroptosis activation markers phospho-RIPK3 (p-RIPK3) and phospho-MLKL (p-MLKL). Cultured MEF cells with indicated CSNK1G2 gene as in (**B**) were treated with indicated stimuli for 4 hr before the cell extracts were prepared and subjected to western blotting analysis as indicated. (**D and E**) Western blotting analysis of RIPK3-associated RIPK1 and CSNK1G2. Immunoprecipitates using an anti-Flag antibody from extracts of MEF-Flag-RIPK3 and MEF (*Csnk1g2*⁻/⁻)-Flag-RIPK3 cells (D) or HeLa-HA-3×Flag-RIPK3-Myc-CSNK1G2(WT and K75A) cells (E) treated with the indicated stimuli for 6 hr were subjected to western blotting analyzing using antibodies as indicated. (**F**) Cell viability measurement of bone marrow-derived macrophages from the wild-type (WT) or *Csnk1g2* knockout mice. Macrophages were isolated from the WT or *Csnk1g2* knockout mice (KO) and treated with the indicated necroptosis stimuli for 12 hr, and the cell viability was measured by Cell-Titer Glo. Trail: TNF-related apoptosis-inducing ligand. LPS: Lipopolysaccharide. Data are mean ± SD of triplicate wells. ***p<0.001. p-values were determined by two-sided unpaired Student's *t*-tests. (**G**) Kaplan–Meier plot of survival of male *Csnk1g2*⁺/⁺(WT), *Csnk1g2*⁻/⁻ (*Csnk1g2* knockout littermates), or Ripk3⁻/

*Figure 1 continued on next page*

*Figure 1 continued*

$^-$ (*Ripk3* gene knockout) mice (n = 10 for each genotype, age: 3 months) injected intraperitoneally with one dose of murine TNF-α (300 μg/kg). Body temperature was measured with a lubricated rectal thermometer. Mice with a temperature below 23°C were euthanized for ethical reasons. Generation of *Csnk1g2*$^{-/-}$ mice.

The online version of this article includes the following figure supplement(s) for figure 1:

**Figure supplement 1.** CSNK1G2 binds and inhibits RIPK3 kinase activity to prevent necroptosis.
**Figure supplement 2.** The kinase activity of CSNK1G2 is required for its binding to RIPK3.
**Figure supplement 3.** Generation of *Csnk1g2*$^{-/-}$ mice.

and 4), indicating that CSNK1G2 suppresses necroptosis by binding to RIPK3 and preventing its recruitment to necrosome.

In addition to MEFs, we investigated the effect of CSNK1G2 in human cells. As shown in *Figure 1E*, expressing a Myc-tagged human CSNK1G2 reduced RIPK1 and RIPK3 interaction as measured by a co-immunoprecipitation (co-IP) experiment (*Figure 1E*, lanes 2 and 4). The kinase-dead mutant of CSNK1G2 (K75A) neither co-IPed with RIPK3 nor decreased RIPK1–RIPK3 interaction (*Figure 1E*, lanes 5 and 6, and *Figure 1—figure supplement 2C*). Moreover, the wild-type CSNK1G2-blocked necroptosis caused by the FKBP-binding small molecule-induced dimerization of an FKBP–F36V–RIPK3 fusion protein (*Li et al., 2020*; *Orozco et al., 2014*), while the kinase-dead CSNK1G2 (K75A) mutants did not, indicating that CSNK1G2 inhibits necroptosis by directly preventing RIPK3 activation (*Figure 1—figure supplement 2D*).

## *Csnk1g2* knockout mice showed accelerated TNF-α-induced systematic sepsis

We subsequently knocked out the *Csnk1g2* gene in mice using guide RNA specifically targeted to exon-2 of the *Csnk1g2* gene (*Figure 1—figure supplement 3A*). The successful deletion of 56 base pairs of exon-2 in the *Csnk1g2* gene caused the deletion of its N-terminus kinase domain and introduced a new premature stop codon in the remaining mRNA (*Figure 1—figure supplement 3A,B and D*). Knockout of the *Csnk1g2* gene resulted in no detection of CSNK1G2 protein in the testis of these animals (*Figure 1—figure supplement 3C*).

Loss of the *Csnk1g2* gene did not affect the development of the knockout mice. However, the bone marrow-derived macrophages (BMDM) from the *Csnk1g2* knockout mice showed enhanced cell death when treated with necroptosis-inducing agents, including T/S/Z, TRAIL/S/Z, or LPS/S/Z (*Figure 1F*) compared to the BMDM from their wild-type littermates. Notably, although the *Csnk1g2* knockout mice looked normal, they died within 18 hr after TNF-α administration, much quicker than their wild-type littermates, indicating that TNF-α-induced systematic sepsis was dramatically enhanced (*Figure 1G*). In contrast, *Ripk3* knockout mice were totally resistant to such a treatment as previously reported (*Newton et al., 2014*; *Figure 1G*).

## CSNK1G2 interaction with RIPK3 requires auto-phosphorylation of its serine 211 and threonine 215 sites

Since the interaction between CSNK1G2 and RIPK3 requires the kinase activity of CSNK1G2, we searched for phosphorylation events on these two proteins that might be required for such an interaction. To this end, we immunoprecipitated CSNK1G2–RIPK3 complex and by mass spectrometry analysis found three clusters of peptides of CSNK1G2 that contained phosphorylated amino acid residues. These residues were: serine 26 and serine 27, serine 211 and threonine 215, and serine 381 (*Figure 2A* and *Figure 2—figure supplement 1A*). We subsequently introduced phosphorylation-resistant mutations in these residues and assessed their effect on RIPK3 activity. We found that only the S211A/T215A mutant lost the ability to block RIPK3 S227 phosphorylation, whereas the other two mutants containing serine 26/serine 27 and serine 381 to alanine mutations still blocked RIPK3 S227 phosphorylation as efficiently as the wild type (*Figure 2B* and *Figure 2—figure supplement 1B*). Consistently, reintroducing S211A/T215A mutant CSNK1G2 to the *Csnk1g2* knockout MEFs did not restored the necroptosis-inhibiting activity of CSNK1G2 (*Figure 2C*). Furthermore, compared to wild-type CSNK1G2 that efficiently bound to RIPK3 and blocked RIPK1–RIPK3 interaction induced by necroptosis inducer TSZ (*Figure 2D*, lanes 1–4), S211A/T215A mutant CSNK1G2 lost its ability to

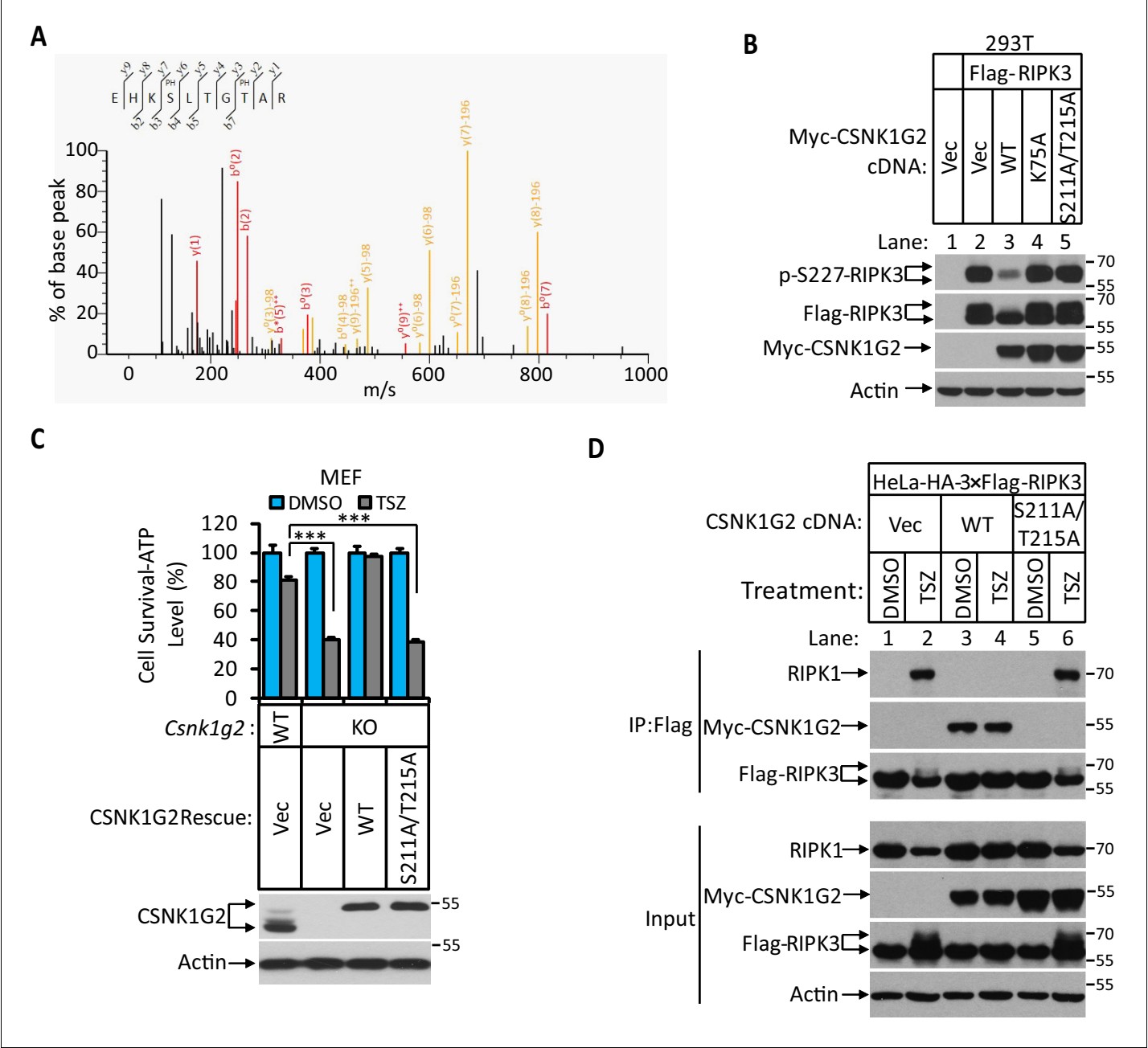

**Figure 2.** Identification and characterization of CSNK1G2 auto-phosphorylation sites. (**A**) MS/MS spectrum of CSNK1G2 phosphorylation sites. The identified phosphorylated peptide to be EHK$_P$SLTG$_P$TAR with S211 and T215 as the phosphorylated amino acid residues. The b- and y-type product ions are indicated in the spectrum. Data are related to those in *Figure 2—figure supplement 1*. (**B**) Effect on RIPK3 auto-phosphorylation by co-expression of the indicated version of CSNK1G2. Cultured 293 T cells were transfected with Flag-tagged RIPK3 cDNA together with indicated Myc-tagged wild-type (WT) CSNK1G2, or kinase-dead mutant K75A, or phosphorylation site-resistant mutant S211A/T215A for 20 hr. The cell extracts were then subjected to western blotting analysis using antibodies against phospho-S227-RIPK3, Flag-RIPK3, and Myc-CSNK1G2 as indicated. (**C**) Top: Cell viability measurement of effect of phosphorylation sites mutant CSNK1G2 on necroptosis. Cultured WT MEF or MEF with their *Csnk1g2* gene knocked out (KO) were transfected with either vector control (Vec) or cDNA encoding WT *Csnk1g2* or phosphorylation sites mutant (S211A/T215A) followed by treatment of DMSO or necroptotic stimuli TSZ as indicated for 12 hr. The cell viability was measured by Cell-titer Glo. Data are mean ± SD of triplicate wells. ***p<0.001. p-values were determined by two-sided unpaired Student's *t*-tests. Bottom: Aliquots of these treated cells were used to make cell extracts for western blotting analysis using an antibody against CSNK1G2 protein. The effect of CSNK1G2 on RIPK1/RIPK3 interaction as measured by co-IP. Cultured HeLa-HA-3×Flag-RIPK3 cells were transfected with either vector control (Vec) or WT or phosphorylation site mutant Myc-CSNK1G2 (S211A/T215A) as indicated. The cells were then treated with DMSO or necroptosis stimuli TSZ for 6 hr. The cell extracts were prepared and subjected

*Figure 2 continued on next page*

*Figure 2 continued*

to immunoprecipitation with an anti-Flag antibody. The extracts (Input) and the immunoprecipitates (IP: Flag) were then subjected to western blotting analysis using antibodies as indicated.

The online version of this article includes the following figure supplement(s) for figure 2:

**Figure supplement 1.** Auto-phosphorylation sites on CSNK1G2.

bind RIPK3 and did not block RIPK1–RIPK3 interaction in response to TSZ (*Figure 2D*, lanes 5 and 6). Moreover, single S211A or T215A mutant showed decreased ability to block RIPK3 kinase activity while the double mutant lost all the inhibitory activity, similar to the K75A kinase-dead mutant (*Figure 2—figure supplement 1C*). These results suggested that the auto-phosphorylation of serine 211 and threonine 215 contributed to the ability of CSNK1G2 to bind and inhibit RIPK3 kinase activity. Not surprisingly, serine 211 and threonine 215 are within the conserved region of CSNK1G2 with amino acid residues between 205 and 240 (human origin) 100% conserved between human, chimpanzee, mouse, and bovine (*Figure 2—figure supplement 1D*).

## Knocking out *Csnk1g2* in testis cells significantly enhanced their necroptosis response

To further study the function of *Csnk1g2* in vivo, we first measured the expression of this protein in mouse tissues by western blotting. As shown in *Figure 3A*, CSNK1G expression is low in the brain, heart, liver, ovary, and intestine (*Figure 3A*, lanes 3, 4, 6, 7, 8, 9, and 10). There was higher CSNK1G2 presence in lung and spleen (*Figure 3A*, lanes 2 and 5). The highest expression was found in the testis (*Figure 3A*, lane 1). Immunohistochemical analysis showed that CSNK1G2 was present in the seminiferous tubules of the testis (*Figure 3B*) and overlapped with that of RIPK3, indicating that, like RIPK3, it is also expressed in the spermatogenic cells and Sertoli cells, two major cell types in the seminiferous tubules that express RIPK3 (*Li et al., 2017*). When primary cells from the testis of *Csnk1g2* knockout mice and their wild-type littermates were treated with necroptosis stimuli TSZ, significantly more cell death was observed in cells from the *Csnk1g2* knockout testis (*Figure 3C*). Additionally, when the endogenous CSNK1G2 was immunoprecipitated from the testis extracts of 3-month-old wild-type mice, the endogenous RIPK3 was co-precipitated, indicating that these two proteins not only co-express in the testis but also interact with each other (*Figure 3D*). In contrast, the same immunoprecipitation experiment using testis extracts from the CSNK1G2 knockout littermates did not precipitate either CSNK1G2 or RIPK3, confirming the specificity of the anti-CSNK1G2 antibody (*Figure 3D*). These findings further indicate that the function of CSNK1G2 in testis is to block necroptosis in the RIPK3-expressing spermatogenic cells and Sertoli cells by directly binding to RIPK3.

To further demonstrate necroptosis suppression activity of CSNK1G2 in seminiferous cells in testis, we knocked out the *Csnk1g2* gene in cell lines derived from spermatocyte (GC-2spd(ts)), or Sertoli cells (15 p-1) and measured their necroptosis response. Similar to what was seen in testis, GC-2spd and 15 p-1 cells expressed both RIPK3 and CSNK1G2. In contrast, a cell line from the testosterone-producing Leydig cells did not express RIPK3, and CSNK1G2 was present at a much lower level compared to the other two cell lines (*Figure 3E* and *Figure 3—figure supplement 1*). Similar to what was observed in primary testis cells, GC-2spd and 15 p-1 with their *Csnk1g2* gene knocked out showed significantly more death compared to their respective parental cells when treated with necroptotic stimuli TSZ (*Figure 3F and G*).

## *Csnk1g2* knockout mice showed accelerated male reproductive system aging compared to their wild-type littermates

When male mice reach more than one-and-a-half years of age, their body weight increases, their seminal vesicles grow several fold in both size and weight, and their seminiferous tubules become empty as spermatogenic and Sertoli cells undergo necroptosis (*Li et al., 2017*). We thus measured these aforementioned physiological features of *Csnk1g2* knockout mice and their wild-type littermates up to 12 months of age. When these mice were at 2 months of age, their body weight, seminal vesicle size, testis, and appearance of seminiferous tubules were indistinguishable (*Figure 4A–F*). However, at 12 months of age, the body weight of *Csnk1g2* knockout mice became significantly

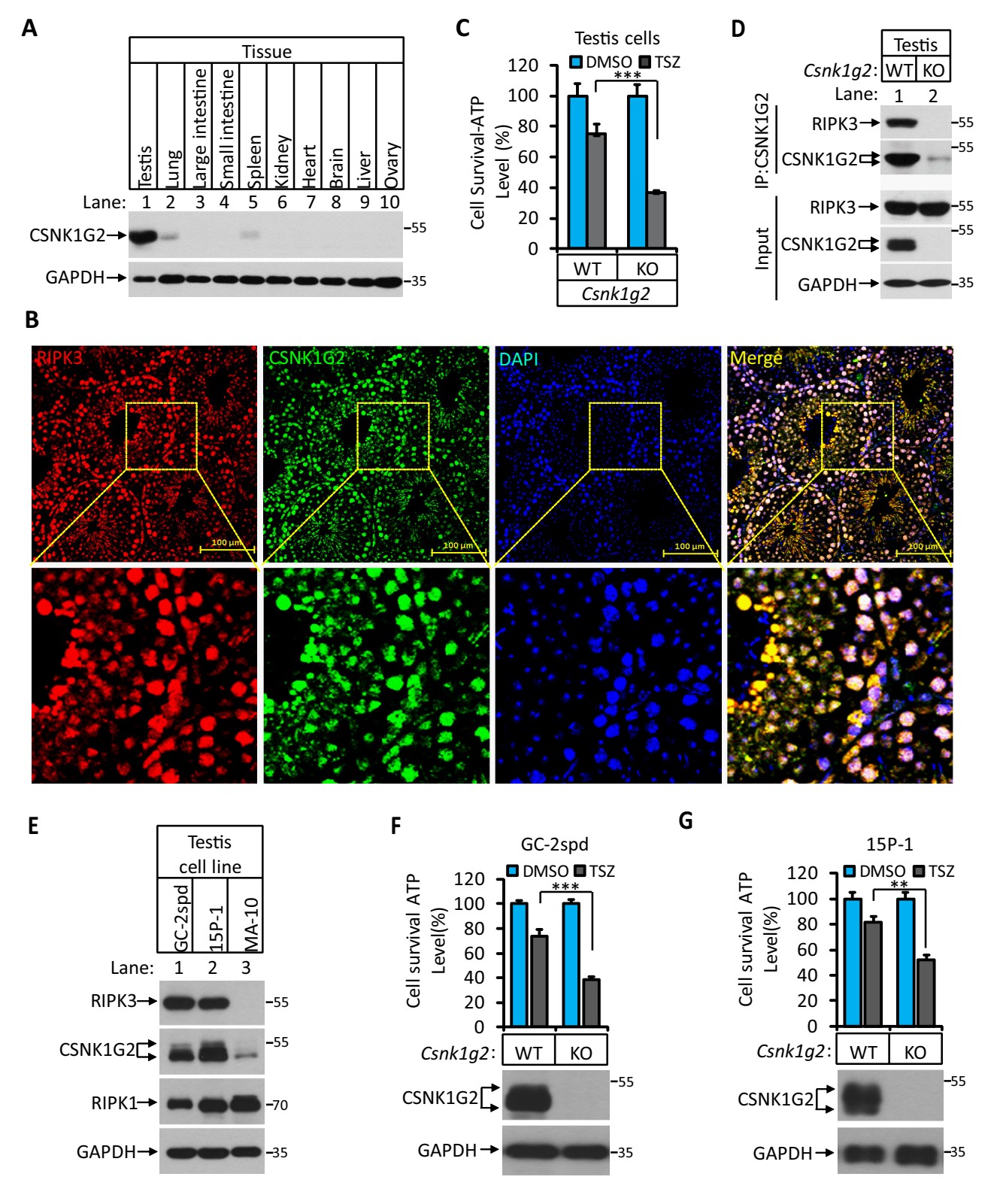

**Figure 3.** CSNK1G2 and RIPK3 are co-expressed in the seminiferous tubules of the mouse testis. (**A**) The expression of CSNK1G2 in mouse testis, lung, large intestine, small intestine, spleen, kidney, heart, brain, liver, and ovary tissues (n = 3, 2 months). The indicated tissue extracts were subjected to western blotting analysis using antibodies against CSNK1G2 and GAPDH as indicated. (**B**) The expression of RIPK3 and CSNK1G2 in mouse testis. The testis sections of 2-month-old wild-type mice (n = 3) were stained sequentially with antibodies against RIPK3 and CSNK1G2 followed by fluorescent-

*Figure 3 continued on next page*

*Figure 3 continued*

conjugated secondary antibody. Counterstaining with DAPI, blue. Scale bar on the upper panel is 100 µm. The areas marked by the yellow boxes on the upper panel were shown in the lower panel. (C) The sensitivity of primary cells from the seminiferous tubules of $Csnk1g2^{-/-}$ and $Csnk1g2^{+/+}$ testis to necroptosis induction. The cells from the seminiferous tubules of 2-month-old littermates with indicated genotype were isolated and cultured in vitro before treated with DMSO or TSZ as indicated for 12 hr. The cell viability was then measured by Cell-Titer Glo. Data are mean ± SD of triplicate wells. ***p<0.001. p-values were determined by two-sided unpaired Student's *t*-tests. (D) Western blotting analysis of CSNK1G2 interaction with RIPK3 in mouse testis. The wild-type and *Csnk1g2* knockout testis (3 months) were ground and resuspended in lysis buffer, homogenized for 30 s with a Paddle Blender. The supernatants were collected, and CSNK1G2 was immunoprecipitated using anti-CSNK1G2 antibody. The immunocomplexes and lysates were analyzed by western blotting using antibodies as indicated. Each group was from a pool of three mice. (E) The expression of RIPK3, CSNK1G2, and RIPK1 proteins in GC-2spd, 15 P-1, and MA-10 cells. The extracts from the indicated cultured cells were subjected to western blotting analysis using antibodies against RIPK3, CSNK1G2, RIPK1, and GAPDH as indicated. (F and G) The effect of CSNK1G2 on necroptosis of GC-2spd and 15 P-1 cells. Cultured parental GC-2spd (f) or 15 P-1 cells (g) (WT), and GC-2spd or 15 P-1 cells with their *Csnk1g2* gene knocked out ($Csnk1g2^{-/-}$) were treated with DMSO or TSZ as indicated for 4 hr. The cell viability was measured by Cell-titer Glo. Data are mean ± SD of triplicate wells. **p<0.01, ***p<0.001. p-values were determined by two-sided unpaired Student's *t*-tests. Bottom, immunoblot of CSNK1G2. Cell extracts from aliquots of these cells were also subjected to western blotting analysis using antibodies against CSNK1G2 and GAPDH as indicated, and the results were shown at the bottom.

The online version of this article includes the following figure supplement(s) for figure 3:

**Figure supplement 1.** Immunofluorescent analysis of RIPK3 and CSNK1G2 expression in GC-2spd, 15 P-1, and MA-10 cells.

higher than their wild-type littermates, reaching up to 45 g on average compared to ~37 g for the wild type (*Figure 4A*). Their seminal vesicles weighted approximately 1 g, a 10-fold increase from when these mice were 2 months old, and about twofold higher than their 12-month-old wild-type littermates (*Figure 4B and C*). The average size and weight of the testis of 12-month-old *Csnk1g2* knockout mice were also significantly smaller than their wild-type littermates, and many of their seminiferous tubules were already empty (*Figure 4D–F*). At 12 months of age, ~31% of *Csnk1g2* knockout seminiferous tubules were empty compared to ~2% empty seminiferous tubules of wild-type littermates (*Figure 4F*).

In addition to the differences in appearance, 12-month-old *Csnk1g2* knockout testis had much more necroptosis activation marker phospho-Serine345-MLKL in their seminiferous tubules. As shown in *Figure 4G and H*, there was prominent staining of phospho-Serine345-MKLK signal in the seminiferous tubules of *Csnk1g2* knockout testis, whereas much less signal was seen in that of wild-type littermates. Consistently, the phospho-serine345-MLKL signal was only detected by western blotting in the testis extracts from 12-month-old *Csnk1g2* knockout mice (*Figure 4I*, lane 4). No signal was observed in the extracts of young mice (2 months), nor from the 12-month-old wild-type littermates, although the protein levels of RIPK1, RIPK3, and MLKL were the same (*Figure 4I*, lanes 1–3).

Finally, we tested the reproductive activity of *Csnk1g2* knockout and their wild-type littermates when they were 2 and 12 months of age. As shown in *Figure 4J*, all twelve 2-month-old male mice, regardless of genotype, impregnated their 10-week-old female partners. In contrast, only one of the twelve 12-month-old *Csnk1g2* knockout mice was able to produce pubs when mated with 10-week-old female mice, whereas 10 of the 12 wild-type littermates produced progenies with their 10-week-old partners.

## Blocking necroptosis by a RIPK1 inhibitor or *Ripk3* gene knockout prevented CSNK1G2-accelerated male reproductive system aging

To test if the accelerated male reproductive system aging in the *Csnk1g2* knockout mice was indeed due to enhanced necroptosis in the testis of these animals, the *Csnk1g2* knockout animals were either fed a RIPK1 kinase inhibitor RIPA-56-containing diet or crossed with *Ripk3* knockout mice to generate *Csnk1g2/Ripk3* double knockout mice. As shown in *Figure 5A–C*, *Figure 5—figure supplement 1A and B*, the signs of aging, including body weight gain, an increase in the size and weight of seminal vesicles, and a decrease in the size of the testis, were all mitigated when the *Csnk1g2* knockout animals were fed with RIPA-56 or their *Ripk3* gene was knockout. All three

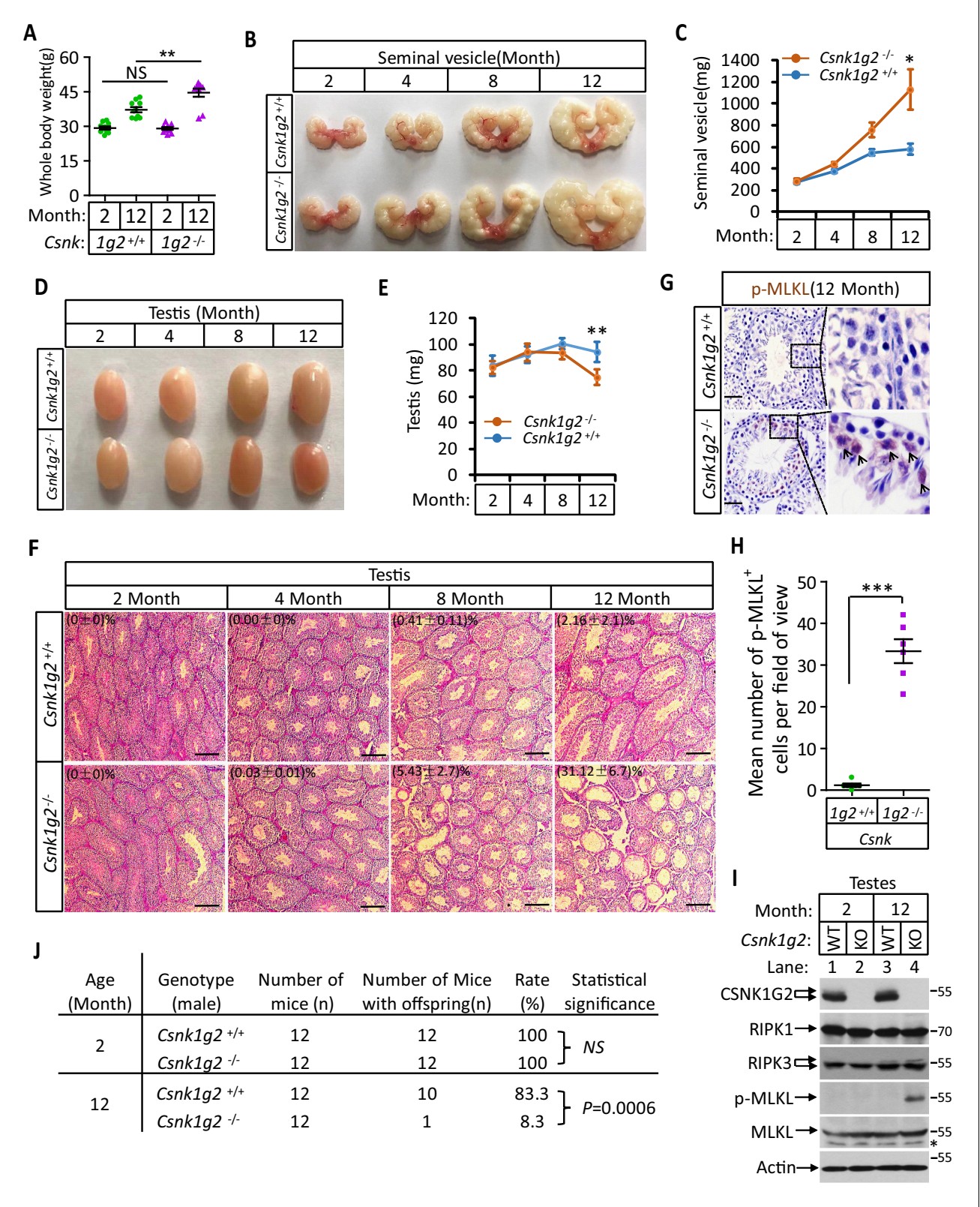

**Figure 4.** Accelerated reproductive system aging in *Csnk1g2⁻/⁻* male mice. (**A**) Body weights of *Csnk1g2⁺/⁺* and *Csnk1g2⁻/⁻* male littermate mice when they were 2 and 12 months old (n = 10 for each genotype). (**B and C**) Macroscopic features (**B**) and weights (**C**) of seminal vesicles from *Csnk1g2⁺/⁺* and *Csnk1g2⁻/⁻* male littermate mice (n = 10 for each genotype) at the indicated ages. (**D and E**) Macroscopic features (**D**) and weights (**E**) of testes from *Csnk1g2⁺/⁺* and *Csnk1g2⁻/⁻* male littermate mice (n = 12 for each genotype) at the indicated ages. (**F**) H&E staining sections of testis from

*Figure 4 continued on next page*

*Figure 4 continued*

*Csnk1g2*$^{+/+}$ and *Csnk1g2*$^{-/-}$ male littermate mice (n = 10 for each genotype) at the indicated ages. The number of empty seminiferous tubules was counted based on H&E staining, and the percentage of empty seminiferous tubules of each group is labeled in the upper left corner of the images. Scale bar, 200 µm. (**G and H**) Immunohistochemical staining (IHC) of testes from *Csnk1g2*$^{+/+}$ and *Csnk1g2*$^{-/-}$ male littermate mice (n = 6 for each genotype) with phospho-MLKL (p-MLKL) antibody in (**G**). p-MLKL positive cells were counted in five fields per testis and quantified in (**H**). Scale bar, 100 µm. (**I**) Western blotting analysis of extracts from phosphate-buffered saline (PBS) perfused testes of *Csnk1g2*$^{+/+}$ (WT) and *Csnk1g2*$^{-/-}$ (KO) male littermate mice of 2 and 12 months of age using antibodies against CSNK1G2, RIPK1, RIPK3, MLKL, and phospho-MLKL (p-MLKL) and β-actin as indicated. The number on the right is markers of molecular weight (kDa). Each group was from a pool of three mice. (**J**) Summary of fertility rates of *Csnk1g2*$^{+/+}$ and *Csnk1g2*$^{-/-}$ male littermate mice (n = 12). Each male mice of 2 or 12 months of age was housed in the same cage with a pair of 10-week-old wild-type female mice for 2 months; females were replaced every 2 weeks. The number of male mice with reproduction capacity was counted. p-values were determined using Fisher's exact tests (unpaired, two-tailed). All quantified data in the figure except (**J**) represent the mean ± SEM. *p<0.05, **p<0.01, ***p<0.001. p-values were determined by two-sided unpaired Student's *t*-tests. NS, not significant.

physiological features remained at the same level as when the animals were young (3 months old) (***Figure 5—figure supplement 1C***).

In addition to these gross physical properties, the number of emptied seminiferous tubules in 12-month-old mice was significantly dropped from about 30% in *Csnk1g2* knockout mice fed with normal chow diet to about 4% in *Csnk1g2* knockout mice fed with RIPA-56-containing diet (***Figure 5D***). The number decreased to about 2%, similar to that of 2-month-old mice, when the mice also had their RIP3 gene knocked out (***Figure 5D***, right panel). The necroptosis activation marker phospho-Serine345-MLKL in the seminiferous tubules of *Csnk1g2* knockout testis was also dropped to almost non-detectable levels in the littermates fed with RIPA-56 or had their *Ripk3* gene knocked out, in contrast to the prominent signal from 12-month-old *Csnk1g2* knockout mice on normal chow (***Figure 5E and F***).

When 12-month-old *Csnk1g2* knockout male mice were tested for their reproductive ability, the loss of reproductive function of these mice was almost completely restored when fed with RIPA-56-containing food or had their *Ripk3* gene knocked out. All ten 12-month-old male *Csnk1g2* knockout mice fed with RIPA-56-containing food produced progenies, and nine of the ten *Csnk1g2/Ripk3* double knockout mice generated pubs when paired with young (10-week-old) female partners, whereas only two of the thirteen *Csnk1g2* knockout littermates on chow diet still produced progenies under the same condition (***Figure 5G***).

The accelerated aging of the mouse male reproductive system observed in 12-month-old *Csnk1g2* knockout mice also manifested in decreased testosterone levels and increased amount of sex hormone-binding globulin (SHBG), pituitary hormone follicle-stimulating hormone (FSH), and luteinizing hormone (LH) (***Figure 5—figure supplement 2E–H***). The change in hormonal levels was mitigated when 12-month-old *Csnk1g2* knockout animals were fed with RIPA-56 or had their *Ripk3* gene knocked out, and all four hormones remained at the same level as when the animals were young (3 months old) (***Figure 5—figure supplement 2A–H***).

## Human testes express CSNK1G2 and showed necroptosis activation marker when old

To test if the observed CSNK1G2 expression in mouse testis also occurred in human testis, we analyzed immunohistochemical sections of human testis surgically removed from young men who suffered from severe testicular torsion. As shown in ***Figure 6A***, CSNK1G2 was expressed in the seminiferous tubules of testis in young men, and the expression of CSNK1G2 completely overlapped with RIPK3. Interestingly, when we examined the testis of both young and older men, we saw that most of the seminiferous tubules in an 82-year-old's testis were empty, whereas the seminiferous tubules of a 30-year-old young man were full of cells surrounding vacuolar center where the mature sperms were present (***Figure 6B and C***). The seminiferous tubules of an 80-year-old testis also had high signal for phospho-Serine358-MLKL, a marker of necroptosis activation (***Figure 6D and E***). No such signal was detected in the seminiferous tubules of a 30-year-old testis (***Figure 6D and E***). These findings indicated that necroptosis-promoted testis aging observed in mice is also conserved in men.

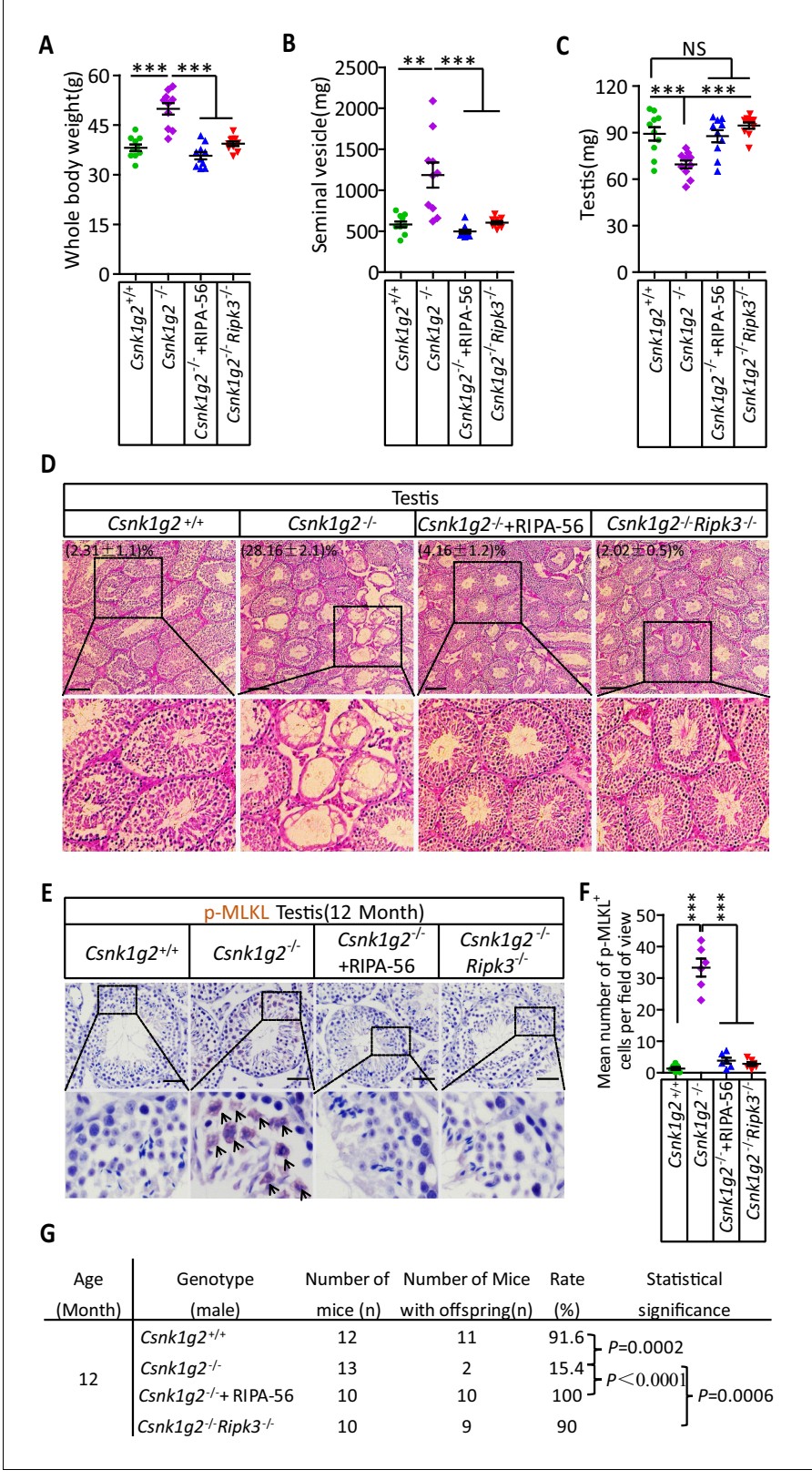

**Figure 5.** Rescuing the accelerated male reproductive system aging of *Csnk1g2* knockout mice with a RIPK1 kinase inhibitor or *Ripk3* knockout. (A) The body weights of 12-month-old *Csnk1g2*[+/+], *Csnk1g2*[−/−], and *Csnk1g2*[−/−] fed with a RIPK1 kinase inhibitor (RIPA-56)-containing diet, and *Csnk1g2*[−/−]*Ripk3*[−/−] male littermate mice (n = 10 for each genotype). (B) The weights of seminal vesicles from 12-month-old *Csnk1g2*[+/+], *Csnk1g2*[−/−], *Csnk1g2*[−/−]
*Figure 5 continued on next page*

*Figure 5 continued*

$^-$+RIPA-56, and $Csnk1g2^{-/-}Ripk3^{-/-}$ male littermate mice (n = 10 for each genotype). (**C**) The weights of testes from 12-month-old $Csnk1g2^{+/+}$, $Csnk1g2^{-/-}$, $Csnk1g2^{-/-}$+RIPA-56, and $Csnk1g2^{-/-}Ripk3^{-/-}$ male littermate mice (n = 10 for each genotype). (**D**) H&E staining of testis sections from 12-month-old $Csnk1g2^{+/+}$, $Csnk1g2^{-/-}$, $Csnk1g2^{-/-}$+RIPA-56, and $Csnk1g2^{-/-}Ripk3^{-/-}$ male littermate mice (n = 10 for each genotype). The number of empty seminiferous tubules was counted based on H&E staining, and the percentage of empty seminiferous tubules was labeled in the upper left corner of the images: scale bar, 200 μm. (**E and F**) IHC staining of testes from 12-month-old $Csnk1g2^{+/+}$, $Csnk1g2^{-/-}$, $Csnk1g2^{-/-}$+RIPA-56, and $Csnk1g2^{-/-}Ripk3^{-/-}$ male littermate mice (n = 8 for each genotype) with an anti-phospho-MLKL (p-MLKL) antibody (**E**). p-MLKL positive cells were counted in five fields per testis and quantified in (**F**). Scale bar, 100 μm. (**G**) Summary of the fertility rates of 12-month-old $Csnk1g2^{+/+}$, $Csnk1g2^{-/-}$, $Csnk1g2^{-/-}$+RIPA-56, and $Csnk1g2^{-/-}Ripk3^{-/-}$ male littermate mice (n = 10 for each genotype). Each male mouse was caged with a pair of 10-week-old wild-type female mice for 2 months; females were replaced every 2 weeks. The number of male mice with reproduction capacity was counted. p-values were determined using Fisher's exact tests (unpaired, two-tailed). $Csnk1g2^{-/-}$+RIPA-56 mice: $Csnk1g2^{-/-}$ male mice were fed with AIN93G or AING3G containing RIPA-56 (RIPA-56: 300 mg/kg) for 10 months started when they were 2 months old in an SPF facility. All quantified data in the figure except (**G**) represent the mean ± SEM. **p<0.01, ***p<0.001. p-values were determined by two-sided unpaired Student's *t*-tests. NS, not significant.

The online version of this article includes the following figure supplement(s) for figure 5:

**Figure supplement 1.** RIPK1 inhibitor-containing diet or double knockout *Ripk3* prevents the appearance of accelerated male reproduction organ aging feature in *Csnk1g2* knockout mice.

**Figure supplement 2.** RIPK1 inhibitor-containing diet or double knockout *Ripk3* prevents the hormonal changes associated with the accelerated male reproduction system aging in *Csnk1g2* knockout mice.

## Discussion

Necroptosis, although not crucial for normal mammalian development, is known to have critical functions in antimicrobial infection and tissue damage response due to the nature of the inflammation-eliciting necrotic cell death that releases damage pattern recognition (DAMP) signals (*Christofferson and Yuan, 2010*; *Vandenabeele et al., 2010*; *Wallach et al., 2016*). It is thus essential that such a danger signal has to be well controlled to prevent accidental death. So far, the most dominant negative regulator identified is active caspase-8, which cleaves and inactivates RIPK1 and RIPK3 kinases, thereby terminating necroptosis (*Günther et al., 2011*; *Kaiser et al., 2011*; *Newton et al., 2019*; *Oberst et al., 2011*). Inhibition or genetic inactivation of caspase-8 activates necroptosis both in vitro and in vivo. Additionally, Ppm1b phosphatase has been proposed to directly dephosphorylate threonine 231 and serine 232 sites of RIPK3 to attenuate the recruitment of MLKL, although the effect is modest at best (*Chen et al., 2015*). Moreover, even after MLKL translocated to the plasma membrane, the MLKL-containing membrane fraction lipid rafts can still be removed from the membrane by flotillin-mediated endocytosis or ESCRT-mediated exocytosis before membrane leakage occurs (*Fan et al., 2019*; *Gong et al., 2017*; *Yoon et al., 2017*). All these regulatory events happen after RIP3 kinase is already activated and thus can only serve as 'emergency brakes'.

Our current report identified CSNK1G2 as an upstream negative regulator of pre-activated RIPK3. CSNK1G2 physically interacts with RIPK3 and prevents RIPK3 from responding to upstream activating signals, be that TNF-α, TRAIL, or LPS. CSNK1G2's function appears analogous to cFlip protein for caspase-8 (*Chang et al., 2002*) or Bcl-2 for Bax/Bak, both of which bind and sequester their pro-death protein targets (*Elmore, 2007*). The binding of RIPK3 by CSNK1G2 requires its autophosphorylation on the serine 211 and threonine 215 sites, possibly a structural requirement to form the CSNK1G2/RIPK3 complex (*Figure 6—figure supplement 1*). This complex formation prevents RIPK3 from being recruited to the necrosome, thus attenuating necroptosis. Since RIPK3 homodimerization is a possible requirement for its activation marked by auto-phosphorylation at serine 227, CSNK1G2, by binding to the monomeric RIPK3, effectively suppresses RIPK3 activation. In addition to CSNK1G2, other members of the casein kinase-1 family, including CSNK1D1 and CSNK1E, seem to be able to bind and suppress RIPK3. These members may regulate necroptosis and other functions of RIPK3 in tissues other than testis, a hypothesis certainly merits future studies.

This CSNK1G2-mediated RIPK3 suppression is particularly important to attenuate necroptosis from occurring in mouse testis, a cell death program that actively promotes male reproductive

system aging (*Figure 6—figure supplement 1*). It is thus not surprising that knockout CSNK1G2 resulted in drastically accelerated aging of the testis. CSNK1G2 is also expressed in human testis, and the necroptosis marker phospho-MLKL was present in the old but not young testis of men. Therefore, the CSNK1G2-accelerated and necroptosis-mediated testis aging phenomenon is evolutionarily conserved between mice and men.

# Materials and methods

## Key resources table

| Reagent type (species) or resource | Designation | Source or reference | Identifiers | Additional information |
|---|---|---|---|---|
| Cell line (*Homo sapiens*) | HEK293T | ATCC | CRL-11268 | Female |
| Cell line (*Homo sapiens*) | HeLa | ATCC | CCL-2 | Female |
| Cell line (*Mus musculus*) | GC-2spd(ts) | ATCC | CRL-2196TM | Male |
| Cell line (*Mus musculus*) | 15 P-1 | ATCC | CRL-2618TM | Male |
| Cell line (*Mus musculus*) | MA-10 | ATCC | CRL-3050TM | Male |
| Cell line (*Homo sapiens*) | HeLa-HA-3×Flag-RIPK3 | Dr. Xiaodong Wang lab at National Institute of Biological Sciences, Beijing | N/A | |
| Cell line (*Homo sapiens*) | HeLa-HA-3×Flag-RIPK3-Myc-CSNK1G2 | Dr. Xiaodong Wang lab at National Institute of Biological Sciences, Beijing | N/A | |
| Cell line (*Homo sapiens*) | HeLa-HA-3×Flag-RIPK3-Myc-CSNK1G2(K75A) | Dr. Xiaodong Wang lab at National Institute of Biological Sciences, Beijing | N/A | |
| Cell line (*Homo sapiens*) | HeLa-HA-3×Flag-RIPK3-Myc-CSNK1G2 (S211A/T215A) | Dr. Xiaodong Wang lab at National Institute of Biological Sciences, Beijing | N/A | |
| Cell line (*Mus musculus*) | NIH3T3-Flag-mRIPK3 | Dr. Xiaodong Wang lab at National Institute of Biological Sciences, Beijing | N/A | |
| Cell line (*Mus musculus*) | NIH3T3-Flag-mRIPK3-Myc-mCSNK1G2 | Dr. Xiaodong Wang lab at National Institute of Biological Sciences, Beijing | N/A | |
| Cell line (*Mus musculus*) | NIH3T3-Flag-mRIPK3-Myc-mCSNK1G2(K75A) | Dr. Xiaodong Wang lab at National Institute of Biological Sciences, Beijing | N/A | |
| Cell line (*Mus musculus*) | MEF($Csnk1g2^{-/-}$) | Dr. Xiaodong Wang lab at National Institute of Biological Sciences, Beijing | N/A | |
| Cell line (*Mus musculus*) | MEF($Csnk1g2^{-/-}$)-Myc-mCSNK1G2 | Dr. Xiaodong Wang lab at National Institute of Biological Sciences, Beijing | N/A | |
| Cell line (*Mus musculus*) | MEF($Csnk1g2^{-/-}$)-Myc-mCSNK1G2(K75A) | Dr. Xiaodong Wang lab at National Institute of Biological Sciences, Beijing | N/A | |
| Cell line (*Mus musculus*) | MEF($Csnk1g2^{-/-}$)-Myc-mCSNK1G2(S211A/T215A) | Dr. Xiaodong Wang lab at National Institute of Biological Sciences, Beijing | N/A | |
| Antibody | Anti-RIPK3 (rabbit polyclonal) | ProSci | Cat# 2283; RRID:AB_203256 | WB (1:1000) |

*Continued on next page*

*Continued*

| Reagent type (species) or resource | Designation | Source or reference | Identifiers | Additional information |
|---|---|---|---|---|
| Antibody | Anti-CSNK1G2 (rabbit monoclonal) | Abcam | Cat# ab238121 | First described in this paper;WB (1:1000) |
| Antibody | Anti-GAPDH-HRP (mouse monoclonal) | MBL | Cat# M171-1; RRID:AB_10699462 | WB (1:20,000) |
| Antibody | Anti-β-Actin-HRP (rabbit polyclonal) | MBL | Cat# PM053-7; RRID:AB_10697035 | WB (1:20,000) |
| Antibody | Anti-Flag-HRP (mouse monoclonal) | Sigma-Aldrich | Cat# A8592; RRID:AB_439702 | WB (1:10,000) |
| Antibody | Anti-RIPK (rabbit monoclonal) | Cell Signaling | Cat# 3493S; RRID:AB_2305314 | WB (1:1000) |
| Antibody | Anti-Myc-HRP (mouse monoclonal) | MBL | Cat# M192-7 | WB (1:2000) |
| Antibody | Anti-Mouse-MLKL (rabbit polyclonal) | ABGENT | Cat# AP14272b; RRID:AB_11134649 | WB (1:1000) |
| Antibody | Anti-Mouse-p-MLKL (rabbit monoclonal) | Abcam | Cat# ab196436; RRID:AB_2687465 | WB (1:1000) |
| Antibody | Anti-Human-p-MLKL (rabbit monoclonal) | Abcam | Cat# ab187091; RRID:AB_2619685 | WB (1:1000) |
| Antibody | Anti-p-S227-RIPK3 (rabbit monoclonal) | Abcam | Cat# ab209384; RRID:AB_2714035 | WB (1:1000) |
| Antibody | Anti-p-S232-RIPK3 (rabbit monoclonal) | Abcam | Cat# ab222302 | WB (1:1000) |
| Antibody | Mouse anti-rabbit IgG (mouse monoclonal) | Cell Signaling | Cat# 5127S; RRID:AB_10892860 | WB (1:2000) |
| Antibody | Donkey anti-mouse, Alexa Fluor 488 (mouse polyclonal) | Thermo Fisher | Cat# A-21202; RRID:AB_141607 | IF (1:500) |
| Antibody | Donkey anti-mouse, Alexa Fluor 555 (mouse polyclonal) | Thermo Fisher | Cat# A-31570; RRID:AB_2536180 | IF (1:500) |
| Antibody | Donkey anti-rabbit, Alexa Fluor 488 (rabbit polyclonal) | Thermo Fisher | Cat# A-21206; RRID:AB_141708 | IF (1:500) |
| Antibody | Donkey anti-rabbit, Alexa Fluor 555 (rabbit polyclonal) | Thermo Fisher | Cat# A-31572 RRID:AB_162543 | IF (1:500) |
| Antibody | Anti-Flag M2 affinity gel | Sigma-Aldrich | A2220 | |
| Antibody | Anti-c-Myc Agarose | Sigma-Aldrich | 20168 | |
| Recombinant DNA reagent (plasmid) | pWPI-Flag-RIPK3 | This paper | N/A | Described in Materials and methods; available upon request |
| Recombinant DNA reagent (plasmid) | pWPI-Flag-mRIPK3 | This paper | N/A | Described in Materials and methods; available upon request |
| Recombinant DNA reagent (plasmid) | pCDNA3.1-Myc-CSNK1G2 | This paper | N/A | Described in Materials and methods; available upon request |
| Recombinant DNA reagent (plasmid) | pCDNA3.1-Myc-CSNK1G2(K75A) | This paper | N/A | Described in Materials and methods; available upon request |
| Recombinant DNA reagent (plasmid) | pCDNA3.1-Myc-CSNK1G2(D165N) | This paper | N/A | Described in Materials and methods; available upon request |

*Continued on next page*

*Continued*

| Reagent type (species) or resource | Designation | Source or reference | Identifiers | Additional information |
|---|---|---|---|---|
| Recombinant DNA reagent (plasmid) | pCDNA3.1-Myc-CSNK1G2(S211A) | This paper | N/A | Described in Materials and methods; available upon request |
| Recombinant DNA reagent (plasmid) | pCDNA3.1-Myc-CSNK1G2(T215A) | This paper | N/A | Described in Materials and methods; available upon request |
| Recombinant DNA reagent (plasmid) | pCDNA3.1-Myc-CSNK1G2 (S211A/T215A) | This paper | N/A | Described in Materials and methods; available upon request |
| Recombinant DNA reagent (plasmid) | pWPI-Myc-CSNK1G2 | This paper | N/A | Described in Materials and methods; available upon request |
| Recombinant DNA reagent (plasmid) | pWPI-Myc-CSNK1G2(K75A) | This paper | N/A | Described in Materials and methods; available upon request |
| Recombinant DNA reagent (plasmid) | pWPI-Myc-CSNK1G2 (S211A/T215A) | This paper | N/A | Described in Materials and methods; available upon request |
| Recombinant DNA reagent (plasmid) | pWPI-Myc-mCSNK1G2 | This paper | N/A | Described in Materials and methods; available upon request |
| Recombinant DNA reagent (plasmid) | pWPI-Myc-mCSNK1G2(K75A) | This paper | N/A | Described in Materials and methods; available upon request |
| Peptide, recombinant protein | Myc peptide | Sigma-Aldrich | 20170 | 1 mg/ml |
| Peptide, recombinant protein | 3xFlag peptide | ChinaPeptides | DYKDHDGDYKDH DIDYKDDDDK | 1 mg/ml |
| Software, algorithm | ImageJ | NIH | N/A | |
| Software, algorithm | GraphPad | Graphpad Software | N/A | |
| Software, algorithm | Prism | Graphpad Software | N/A | |
| Software, algorithm | Nikon A1-R | Nikon | https://www.nikoninstruments.com/Products/Confocal-Microscopes/A1R-HD | |

## Mice

The *Csnk1g2* knockout mice were generated using the CRISPR-Cas9 system (*Figure 1—figure supplement 3*). The *Ripk3*$^{-/-}$ (C57BL/6NCrl strain) were kept in our lab (*He et al., 2009*). *Csnk1g2*$^{+/-}$*Ripk3*$^{+/-}$ mice were produced by mating *Csnk1g2*$^{-/-}$ males with *Ripk3*$^{-/-}$ females. *Csnk1g2*$^{-/-}$*Ripk3*$^{-/-}$ mice were produced by mating *Csnk1g2*$^{+/-}$*Ripk3*$^{+/-}$ males with *Csnk1g2*$^{+/-}$*Ripk3*$^{+/-}$ females. The primers used for genotyping are listed below.

*Csnk1g2*-KO-F: AGGTTTCGCACTCGGATCTCACG;
*Csnk1g2*-KO-R: CCCCGAAGTTCCCACAGCCTATC;
*Ripk3*-KO-F: CAGTGGGACTTCGTGTCCG;
*Ripk3*-KO-R: CAAGCTGTGTAGGTAGCACATC.

## Mice husbandry

Mice were group-housed in a 12 hr light/dark (light between 08:00 and 20:00) in a temperature-controlled room (21.1 ± 1℃) at the National Institute of Biological Sciences with free access to water. The ages of mice are indicated in the figure, figure legends, or methods. All animal experiments

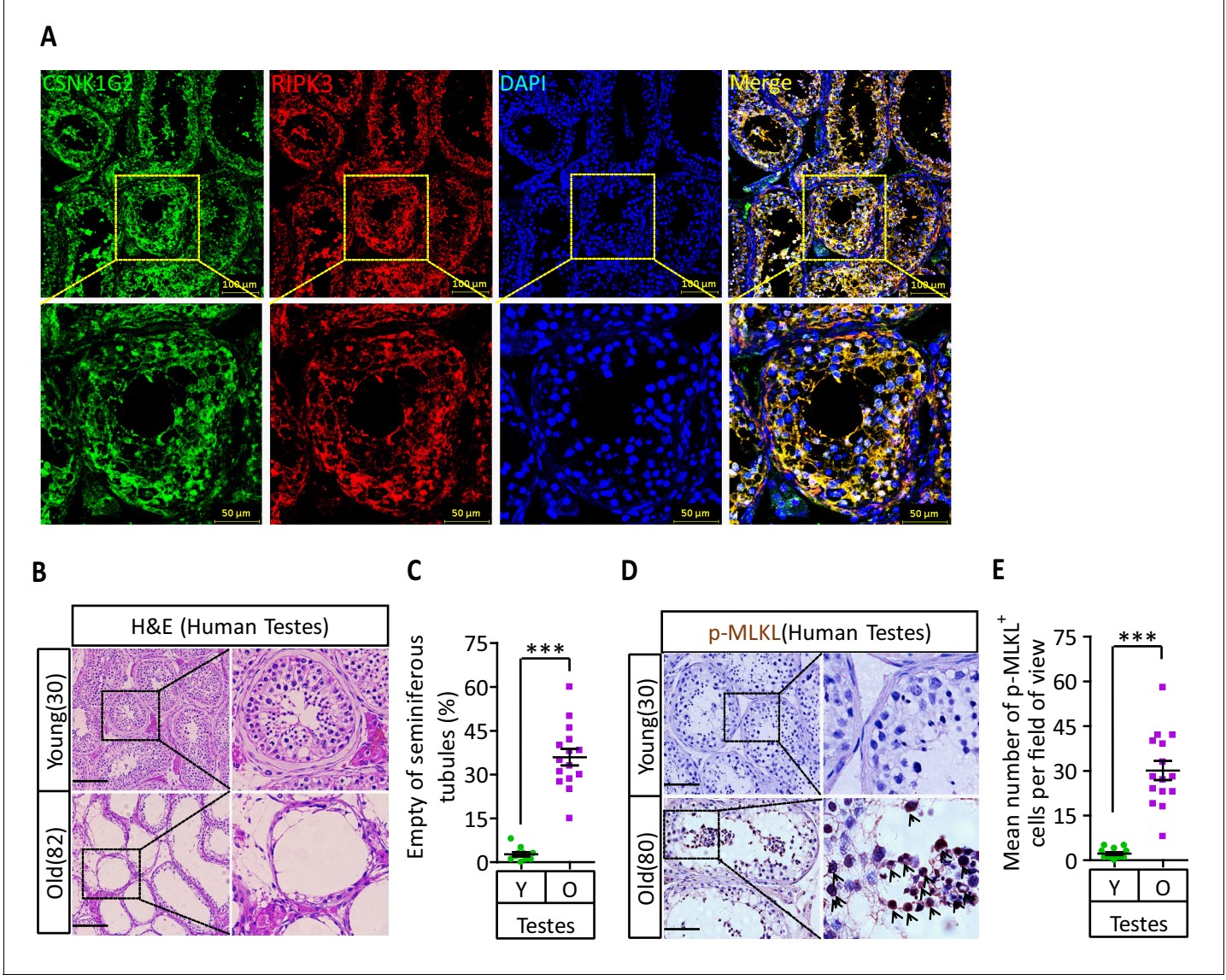

**Figure 6.** CSNK1G2 expression and necroptosis activation marker phosphor-Serine358-MLKL in human testes. (**A**) Expression of CSNK1G2 and RIPK3 in human testes. Sections from a testis sample of a 30-year-old human patient were stained sequentially with antibodies against CSNK1G2 and RIPK3 as indicated followed by green or red fluorescent-conjugated secondary antibodies as indicated. Counterstaining with DAPI, blue. Scale bar, 50/100 μm. Yellow boxes in the upper panels were shown in the lower panels. The experiment was repeated three times with three different patients. (**B and C**) H&E staining of testes from young and old men. Young man testes (25–30 years, n = 10; from testicular torsion necrosis patients) and old man testes (80–89 years, n = 15; from prostate cancer patients) were sectioned and stained with H&E in (**B**). The number of empty seminiferous tubules was counted based on H&E staining and quantification in (**C**); empty seminiferous tubules were counted in five fields per testis. Scale bar, 100 μm. (**D and E**) IHC of testes from young and old men with phosphor-Serine358-MLKL antibody (p-MLKL). Young man testes (25–30 years, n = 10; from testicular torsion necrosis patients) and old man testes (80–89 years, n = 15; from prostate cancer patients) were sectioned and stained with an antibody against phosphor-Serine358-MLKL antibody (**D**). p-MLKL[+] cells were counted in five fields per testis and quantification in (**E**). Scale bar, 100 μm. All quantified data in the figure represent the mean ± SEM. ***p<0.001. p-values were determined by two-sided unpaired Student's *t*-tests.
The online version of this article includes the following figure supplement(s) for figure 6:

**Figure supplement 1.** CSNK1G2 suppresses necroptosis-promoted testis aging by binding and inhibiting RIPK3.

were conducted following the Ministry of Health national guidelines for the housing and care of laboratory animals and were performed in accordance with institutional regulations after review and approval by the Institutional Animal Care and Use Committee at the National Institute of Biological Sciences, Beijing.

## Human tissues

The research involving human tissue samples dissected from adult human testicular torsion necrosis and prostate cancer patients (n = 10, 25–30 years, testicular torsion necrosis patients; n = 25, 80–89 years, prostate cancer patients) were kindly provided by Shanghai Renji Hospital, China, and were snap-frozen in liquid nitrogen and stored at −80°C. Tissues were cut into appropriately sized pieces and placed in formalin for preservation. After several days of formalin fixation at room temperature, tissue fragments were transferred to 70% ethanol and stored at 4°C.

The medical ethics committee of the National Institute of Biological Sciences, Beijing, China, approved the study.

## Ethics

Animal experimentation: Animal care and use followed the institutional guidelines of the National Institute of Biological Sciences (NIBS), Beijing (Approval ID: NIBSLuoM15C) and the Regulations for the Administration of Affairs Concerning Experimental Animals of China.

Constructs psPAX2 and pMD2.G were kept in our lab. Full-length human RIPK3 and mouse Ripk3 cDNA were kept in our lab and subcloned into the pWPI (GFP-tagged) and pCDNA3.1 vector to generate pWPI-Flag-RIPK3, pWPI-Flag-mRIPK3, and pCDNA3.1-HA-3×Flag-RIPK3 constructs. Full-length cDNA for human CSNK1A1, CSNK1A1-L, CSNK1D1, CSNK1D2, CSNK1G1, CSNK1G2, CSNK1G3, CSNK1E, CSNK2A1, CSNK2A2, and CSNK2B were RCR-amplified from cDNA library (Sigma, USA) using KOD polymerase (TOYOBO, China) and subcloned into pcDNA3.1 vector. Quick-change Site-Directed Mutagenesis Kit was used to generate pWPI-Myc-CSNK1G2(K75A)/CSNK1G2 (D165N)/CSNK1G2(S211A)/CSNK1G2(T215A)/CSNK1G2(S211A/T215A) and pWPI-Myc-mCSNK1G2/ CSNK1G2(K75A)/CSNK1G2(S211A)/CSNK1G2 (T215A)/CSNK1G2 (S211A/T215A) constructs.

The gRNAs for targeting *Csnk1g2* (*Figure 1—figure supplement 3A*) were designed and were cloned into the gRNA-Cas9 expression plasmid pX458-GFP to generate pX458-GFP-mCSNK1G2 construct.

## Cells

All cells were cultured at 37°C with 5% $CO_2$. All cell lines were confirmed to be mycoplasma free on a weekly basis using a PCR-based assay with primers: 5′-GGGAGCAAACAGGATTAGATACCCT-3′ and 5′- TGCACCATCTGTCACTCTGTTAACCTC-3′. All cell lines were cultured as follows: HT29 cells were obtained from ATCC and cultured in McCoy's 5A culture medium(GIBCO, 16600082, USA). HEK293T (293T), HeLa, GC-2spd(ts), and 15 P-1 cells were obtained from ATCC and cultured in DMEM (GIBCO, C11965500BT, USA). NIH3T3-Flag-RIPK3 (NIH3T3 stably transfected with Flag-RIPK3 fused to FKBP-F36V), MEFs, MEF (*Csnk1g2*$^{−/−}$), and HeLa-HA-3×Flag-RIPK3 (*Sun et al., 2012*) cells were cultured in DMEM (GIBCO, C11965500BT, USA). NIH3T3-Flag-RIPK3, HeLa-HA-3×Flag-RIPK3, and MEF (*Csnk1g2*$^{−/−}$) cells were infected with virus encoding Myc-CSNK1G2 (WT, K75A, and S211A/T215A) or Myc-mCSNK1G2 (WT, K75A, S211A, T215A, and S211A/T215A), and GFP-positive live cells were sorted to establish the NIH3T3-Flag-RIPk3-Myc-CSNK1G2(WT, K75A), MEF (*Csnk1g2*$^{−/−}$)-Myc-CSNK1G2(WT and K75A), and HeLa-HA-3×Flag-RIPK3-Myc-CSNK1G2 (WT and K75A) cell lines. All media were supplemented with 10% FBS (Thermo Fisher, USA) and 100 units/ml penicillin/streptomycin (Thermo Fisher, USA). MA-10 was obtained from ATCC and cultured in DMEM:F12 (GIBCO, USA, additional 20 mM HEPES, horse serum to a final concentration of 15%).

## Isolation of cells from testes seminiferous tubules

Testes from 8-week-old mice were collected using a previously reported protocol (*Chang et al., 2011*; *Li et al., 2017*). Briefly, a testis was placed in Enriched DMEM:F12 (GIBCO, USA) media and placed on ice. After removal of the tunica albuginea of a testis, the seminiferous tubules were dissociated and transferred immediately into 10 ml of protocol enzymatic solution 1. Tubules were incubated for 15–20 min at 35°C in a shaking water bath at 80 oscillations (osc)/min and were then layered over 40 ml of 5% Percoll/95% 1× Hank's balanced salt solution in a 50 ml conical tube and allowed to settle for 20 min. Leydig cells were removed from the top 35 ml of the total volume of Percoll. The bottom 5 ml of Percoll was transferred to a fresh 50 ml conical tube containing 10 ml enzymatic solution 2. Tubules were incubated for 20 min at 35°C and 80 osc/min. After incubation, 3 ml of charcoal-stripped FBS was immediately added to halt the digestion. The fraction

was immediately centrifuged at 500 × g at 4℃ for 10 min. Pellets were resuspended in PBS and washed three times, and then cultured in DMEM:F12 (15% FBS) medium at 37℃.

## Cell survival assay

Cell survival assay was performed using Cell Titer-Glo Luminescent Cell Viability Assay kit. A Cell Titer-Glo assay (Promega, G7570, USA) was performed according to the manufacturer's instructions. Luminescence was recorded with a Tecan GENios Pro plate reader.

## ELISA

Mice were killed and blood was clotted for 2 hr at room temperature before centrifugation at approximately 1000 × g for 20 min. Mice blood sera were collected and assayed immediately or was stored as sample aliquots at −20℃. The testosterone/FSH/LH levels were measured with ELISA kits (BIOMATIK, EKU07605, EKU04284, EKU05693, USA); the SHBG level was measured with an ELISA kit (INSTRUCTION MANUAL, SEA396Mu, USA). The ELISA assays were performed according to the manufacturer's instructions.

## CRISPR/Cas9 knockout cells

Four micrograms of pX458-GFP-mCSNK1G2 plasmid was transfected into $1 \times 10^7$ MEF cells using the Transfection Reagent (FuGENEHD, E2311, USA) by following the manufacturer's instructions. Three days after the transfection, GFP-positive live cells were sorted into single clones by using a BD FACSArial cell sorter. The single clones were cultured into 96-well plates for another 10–14 days or longer, depending upon the cell growth rate. The anti-CSNK1G2 immunoblotting was used to screen for the MEF ($Csnk1g2^{−/−}$) clones. Genome type of the knockout cells was determined by DNA sequencing.

## Western blotting

Cell pellet samples were collected and resuspended in lysis buffer (100 mM Tris-HCl, pH 7.4, 100 mM NaCl, 10% glycerol, 1% Triton X-100, 2 mM EDTA, Roche complete protease inhibitor set, and Sigma phosphatase inhibitor set), incubated on ice for 30 min, and centrifuged at 20,000 × g for 30 min. The supernatants were collected for western blotting. Testis or other tissues were ground and resuspended in lysis buffer, homogenized for 30 s with a Paddle Blender (Prima, PB100, UK), incubated on ice for 30 min, and centrifuged at 20,000 × g for 30 min. The supernatants were collected for western blotting.

## Immunoprecipitation

The cells were cultured on 15 cm dishes and grown to confluence. Cells at 70% confluence and subjected to indicated treatment for the appropriate time according to different experiments. Then cells were washed once with PBS and harvested by scraping and centrifugation at 800 × g for 5 min. The harvested cells were washed with PBS and lysed for 30 min on ice in the lysis buffer (100 mM Tris-HCl, pH 7.4, 100 mM NaCl, 10% glycerol, 1% Triton X-100, 2 mM EDTA, Roche complete protease inhibitor set, and Sigma phosphatase inhibitor set). Cell lysates were then spun down at 12,000 × g for 20 min. The soluble fraction was collected, and the protein concentration was determined by Bradford assay. Cell extracted was mixed with anti-Flag/Myc affinity gel (Sigma-Aldrich, A2220, A7470, USA) in a ratio of 1 mg of extract per 30 µl of agarose. After overnight rocking at 4℃, the beads were pelleted at 2500 × g for 3 min and washed with lysis buffer three times. The beads were then eluted with 0.5 mg/ml of the corresponding antigenic peptide for 6 hr or directly boiled in 1× SDS loading buffer (125 mM Tris, pH 6.8, 2% 2-mercaptoethanol, 3% SDS, 10% glycerol, and 0.01% bromophenol blue).

## Harvesting of tissues

Animals were killed and perfused with PBS, followed by 4% paraformaldehyde. Major organs were removed, cut into appropriately sized pieces, and either flash-frozen in liquid nitrogen and stored at −80℃ or placed in 4% paraformaldehyde for preservation. After several days of 4% paraformaldehyde fixation at room temperature, tissue fragments were transferred to 70% ethanol and stored at

4°C. Blood was collected by cardiac puncture and was allowed to coagulate for the preparation of serum.

## Immunohistochemistry and immunofluorescence

Paraffin-embedded specimens were sectioned to a 5 μm thickness and were then deparaffinized, rehydrated, and stained with hematoxylin and eosin (H&E) using standard protocols. For the preparation of the immunohistochemistry samples, sections were dewaxed, incubated in boiling citrate buffer solution for 15 min in plastic dishes, and subsequently allowed to cool down to room temperature over 3 hr. Endogenous peroxidase activity was blocked by immersing the slides in hydrogen peroxide buffer (10%, Sinopharm Chemical Reagent, China) for 15 min at room temperature and were then washed with PBS. Blocking buffer (1% bovine serum albumin in PBS, China) was added, and the slides were incubated for 2 hr at room temperature. Primary antibody against p-mMLKL or p-MLKL was incubated overnight at 4°C in PBS. After three washes with PBS, slides were incubated with secondary antibody (polymer-horseradish-peroxidase-labeled anti-rabbit, Sigma, USA) in PBS. After a further three washes, slides were analyzed using a diaminobutyric acid substrate kit (Thermo Fisher, USA). Slides were counterstained with hematoxylin and mounted in neutral balsam medium (Sinopharm Chemical, China).

Immunohistochemistry analysis for RIPK3 or CSNK1G2 was performed using an antibody against RIPK3 and CSNK1G2. Primary antibody against RIPK3 was incubated overnight at 4°C in PBS. After three washes with PBS, slides were incubated with DyLight-555 conjugated donkey anti-rabbi/mouse secondary antibodies (Life, USA) in PBS for 8 hr at 4°C. After a further three washes, slides were incubated with CSNK1G2 antibody overnight at 4°C in PBS. After a further three washes, slides were incubated with DyLight-488 conjugated donkey anti-mouse/rabbit secondary antibodies (Life, USA) for 2 hr at room temperature in PBS. After a further three washes in PBS, the cell nuclei were then counterstained with DAPI (Invitrogen, USA) in PBS. Fluorescence microscopy was performed using a Nikon A1-R confocal microscope.

## Mating and fertility tests

$Csnk1g2^{+/+}$, $Csnk1g2^{-/-}$, $Csnk1g2^{-/-}$+RIPA-56, and $Csnk1g2^{-/-}Ripk3^{-/-}$ male littermate mice were housed in an SPF barrier facility. To score vaginal patency, mice were examined daily from weaning until vaginal opening was observed. The fertility rate of males was determined via a standard method (*Cooke and Saunders, 2002*; *Hofmann et al., 2015*; *Li et al., 2017*) by mating a male with a series of pairs of 10-week-old wild-type females for 2 months; females were replaced every 2 weeks (females were either from our colony or purchased from Vital River Laboratory Co(C57BL/6NCrl)). Each litter was assessed from the date of the birth of pups; when pups were born but did not survive, we counted and recorded the number dead pups; for females that did not produce offspring, the number of pups was recorded as '0' (did not produce a litter with a proven breeder male for a period of 2 months). The number of male mice with reproductive capacity was recorded.

## RIPA-56 feeding experiment

RIPA-56 in the AIN93G (LAD3001G, China) at 300 mg/kg was produced based on the Trophic Animal Feed High-tech Co. protocol (*Li et al., 2017*). Cohorts of 2-month-old $Csnk1g2^{-/-}$ male mice were fed with AIN93G or AING3G containing RIPA-56 (RIPA-56: 300 mg/kg) for 10 months in an SPF facility; each male mouse was then mated with four 10-week-old wild-type female mice successively. The number of male mice with reproductive capacity was recorded.

## Mass spectrometry and data analysis

A total of 293 T cells were transfected with Myc-tagged CSNK1G2 (WT and K75) for 24 hr. Then the cell extracts were prepared and used for immunoprecipitation with an anti-Myc antibody. The immunoprecipitates were washed three times with lysis buffer. The beads were then eluted with 0.5 mg/ml of the corresponding antigenic peptide for 6 hr or directly boiled in 1 × SDS loading buffer and subjected to SDS-PAGE. CSNK1G2 bands were excised from SDS-PAGE gel and then dissolved in 2M urea, 50 mM ammonium bicarbonate, pH 8.0, and reduced in 2 mM DTT at 56°C for 30 min followed by alkylation in 10 mM iodoacetamide at dark for 1 hr. Then the protein was digested with sequencing grade modified trypsin (Promega, USA) (1:40 enzyme to total protein) at 37°C overnight.

The tryptic peptides were separated by an analytical capillary column (50 μm × 15 cm) packed with 5 μm spherical C18 reversed phase material (YMC, Kyoyo, Japan). A Waters nanoAcquity UPLC system (Waters, Milford, USA) was used to generate the following HPLC gradient: 0–30% B in 40 min and 30–70% B in 15 min (A = 0.1% formic acid in water, B = 0.1% formic acid in acetonitrile). The eluted peptides were sprayed into a LTQ Orbitrap Velos mass spectrometer (ThermoFisher Scientific, San Jose, CA, USA) equipped with a nano-ESI ion source. The mass spectrometer was operated in data-dependent mode with one MS scan followed by four CID (collision-induced dissociation) and four HCD (high-energy collisional dissociation) MS/MS scans for each cycle. Database searches were performed on an in-house Mascot server (Matrix Science Ltd, London, UK) against human CSNK1G2 protein sequence. The search parameters are: seven ppm mass tolerance for precursor ions; 0.5 Da mass tolerance for product ions; three missed cleavage sites were allowed for trypsin digestion, and the following variable modifications were included: oxidation on methionine, cysteine carbamidomethylation, and serine, threonine, and tyrosine phosphorylation.

## Statistical analysis

All results are representative of three independent experiments. Statistical tests were used for every type of analysis. The data meet the assumptions of the statistical tests described for each figure. Results are expressed as the mean ± SEM or SD. Differences between experimental groups were assessed for significance using a two-tailed unpaired Student's $t$-test using GraphPad prism6 and Microsoft Excel 2017. Fertility rate was assessed for significance using Fisher's exact test (unpaired, two-tailed) using GraphPad prism6 software. The $*p < 0.05$, $**p < 0.01$, and $***p < 0.001$ levels were considered significant. NS, not significant.

## Acknowledgements

We thank Mr. Alex Wang for critically reading and editing the manuscript. This work was supported by institutional grants from the Chinese Ministry of Science and Technology and Beijing Municipal Commission of Science and Technology. The funders had no role in study design, data collection, and interpretation, or the decision to submit the work for publication.

# Additional information

### Funding

| Funder | Grant reference number | Author |
|---|---|---|
| Ministry of Science and Technology of the People's Republic of China | Institutional | Xiaodong Wang |
| Beijing Municipal Commission of Education | Institutional | Xiaodong Wang |

The funders had no role in study design, data collection and interpretation, or the decision to submit the work for publication.

### Author contributions

Dianrong Li, Conceptualization, Data curation, Formal analysis, Investigation, Methodology, Project administration, Writing - review and editing; Youwei Ai, Conceptualization, Data curation, Formal analysis, Investigation, Methodology, Project administration; Jia Guo, Data curation, Formal analysis, Investigation; Baijun Dong, Resources; Lin Li, Data curation, Formal analysis; Gaihong Cai, Data curation, Investigation; She Chen, Resources, Investigation, Methodology; Dan Xu, Resources, Data curation; Fengchao Wang, Resources, Data curation, Formal analysis; Xiaodong Wang, Conceptualization, Data curation, Formal analysis, Supervision, Funding acquisition, Writing - original draft, Project administration, Writing - review and editing

## Author ORCIDs

Dianrong Li (iD) https://orcid.org/0000-0002-5564-3033
Xiaodong Wang (iD) https://orcid.org/0000-0001-9885-356X

## Ethics

Human subjects: This work used human surgical samples for immunohistochemical analysis. The protocool was approved by the institutional review board for usage of human samples.
Animal experimentation: This study was peformed in strict accordance with NIBS insitutional guidance of animal use and the protocool was approved by the institutional animal care and use committee. ID: NIBSLuoM15C.

## Decision letter and Author response

Decision letter https://doi.org/10.7554/eLife.61564.sa1
Author response https://doi.org/10.7554/eLife.61564.sa2

# Additional files

## Supplementary files

• Transparent reporting form

## Data availability

All data generated or analysed during this study are included in the manuscript and supporting files.

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
