## [Decision Letter]

**Acceptance summary:**

Your work provides key evidence that a casein kinase family member is a major negative regulator of RIP3-mediated necroptosis and plays a crucial role in male reproductive aging. It also demonstrates that human testes exhibit increased necroptosis with age, suggesting that necroptosis-driven aging in the male reproductive system is conserved between mice and humans.

**Decision letter after peer review:**

Thank you for submitting your article "Casein Kinase 1G2 Suppresses Necroptosis-Promoted Testis Aging by Inhibiting Receptor-Interacting Kinase 3" for consideration by *eLife*. Your article has been reviewed by three peer reviewers, and the evaluation has been overseen by a Reviewing Editor and Jessica Tyler as the Senior Editor. The following individual involved in review of your submission has agreed to reveal their identity: Meng C Wang (Reviewer #2).

The reviewers have discussed the reviews with one another and the Reviewing Editor has drafted this decision to help you prepare a revised submission.

Summary:

RIP3 kinase is a crucial regulator of necroptosis, and the role of necroptosis in aging remains poorly understood. In this manuscript, the authors have identified casein kinase 1 family members as RIP3 kinase-interacting partners, and demonstrated that the casein kinase CSNK1G2 inhibits necroptosis by binding to and inhibiting RIP3. A mouse knockout of CSNK1G2 display a premature testis aging phenotype, including elevated levels of necroptosis in the testis and reduced fertility, along with reduced lifespan. Importantly, the authors confirmed that these changes in the male reproductive system are due to increased necroptosis, and human testes exhibit increased necroptosis with increasing age. These results led the authors to conclude that CSNK1G2 restricts RIP3-dependent necroptosis in the testis to prevent premature testicular aging.

Collectively, the study is well done and convincing, and the manuscript discovers a novel kinase that regulates the activity of RIP3 and to play a crucial role in male reproductive aging, and also reveals an interesting link between necroptosis and reproductive aging. However, reviewers noted several issues that need to be addressed.

Essential revisions:

1) The authors demonstrate the interaction between RIP3 and CSNK1G2 using overexpression assays. To strengthen their conclusions, the interaction between endogenous RIP3 and CSNK1G2 should be confirmed and validated in untransfected cells or tissue.

2) The role of other CSNK isoforms in RIP3 interactions and necroptosis inhibition is unclear. Mass spectrometry results in Figure 1—figure supplement 1 showed that CSNK1A1, CSNK1D and CSNK1E were pulled down as RIP3-interacting partners. However, CSNK1A did not inhibit RIP3 phosphorylation. Why was that the case? Why was CSNK1G2 not identified in the mass spectrometry experiment? Are any of the other CSNK isoforms expressed in the testis and if so, can they potentially compensate for the loss of CSNK1G2 function in the knockout mice? Since multiple CSNK isoforms could interact with RIP3, do they inhibit RIP3 activation via similar mechanisms in a cell type-specific manner?

3) The authors show that CSNK deficiency impacts testicular aging and mating performance in older mice. Does CSNK1G2 expression, localization, or activity change with age in wild type mouse or human testes?

4) Additional detailed fertility data in Figures 4J and 5G would make the author's argument more convincing. Mice at 12-month of age typically show drastically reduced breeding performance, so it is surprising that 12 of out 12 old wild type male mice gave rise to offspring. Was the breeding performance scored over a defined period of time (e.g. within a month)? Another indication for breeding performance is the size of the litter. If the authors have this data from the original breeding experiment, they should be included in the table in Figures 4J and 5G.

5) In all figures showing testis histology (3B, 4F, 5D), the panels are too small to evaluate the histology. Higher resolution and larger images are necessary to clearly identify the cell types expressing RIP3 and CSNK1G2 and interpret the testis phenotype.

6) At several places in the manuscript, please revise the text to clarify or substantiate conclusions:

– In Figure 3B (subsection “Knocking out CSNK1G2 in Testis Cells Significantly Enhanced Their Necroptosis Response”), the authors report expression of CSNK1G2 and RIP3 in spermatogonial stem cells and Sertoli cells, but the (low-resolution) image appears to show expression in multiple spermatogenic cells. Spermatogonial stem cells cannot be identified without specific co-staining; a hematoxylin+IHC serial section image alongside the IF images would be necessary to identify the cell types, and would make the figure easier to interpret. Figure 3C also claims an effect in spermatogonial stem cells and Sertoli cells, but does not provide support for which cell types are being tested. GC2 cells used in Figure 3D-F are derived from spermatocytes (meiotic germ cells), not spermatogonia (pre-meiotic germ cells). These issues could be addressed without affecting the authors' main message by changing "spermatogonium cells" to "spermatogenic cells" throughout the text, to encompass all stages of spermatogenesis.

– In Figure 4C, the CSNK1G2 KO mice shown in blue have decreased weight compared to WT. Based on the description in the text and images in Figure 4B, the colors might be mistakenly switched between CSNK1G2^-/-^ and ^+/+^. In Figure 4E, the WT and KO labels appear to be swapped.

– The authors should provide numbers and p-values for Figure 1G, and numbers (p-values already provided) for Figures 4A, C, E, H, 5A-C, F, 6C, E, Figure 5—figure supplement 1C, and Figure 5—figure supplement 2A-H.

– In Figure 1A, with WT CSNK1G2, Flag-RIP3 shows up as one band, while with the mutant forms, Flag-RIP3 shows up as several bands. Are those high-molecular weight bands the phosphorylated forms, or are there other PTMs of RIP3 that are also affected by CSNK1G2?

– MEFs express endogenous CSNK1G2 (Figure 1C). Reconstitution of CSNK1G2 in the KO MEFs inhibited RIP3 phosphorylation (Figure 1C). The level of CSNK1G2 expression under this condition was similar to endogenous CSNK1G2 expression in wild type MEFs (Figure 1C, compare lanes 5 and 7). Yet, over-expression in KO MEFs had a stronger effect on RIP3 phosphorylation. Do the authors have an explanation for this?

– Throughout the manuscript, a single time point was chosen to demonstrate the effect of CSNK1G2 on RIP3 activation. The effect was not "all or none", suggesting that CSNK1G2 might affect the kinetics of necrosome formation and necroptosis. It would be helpful to discuss how CSNK1G2 might affect the kinetics of the response; this could provide key insights into the mechanism of CSNK1G2 on necroptosis regulation.

– CSNK1G2 binding to RIP3 seemed to decrease upon TSZ stimulation (Figure 1D). Was this due to de-phosphorylation of CSNK1G2?

---

## [Author Response]

Essential revisions:1) The authors demonstrate the interaction between RIP3 and CSNK1G2 using overexpression assays. To strengthen their conclusions, the interaction between endogenous RIP3 and CSNK1G2 should be confirmed and validated in untransfected cells or tissue.

We agree with the reviewers’ comments that the interaction between the endogenous RIP3 and CSNK1G2 should be confirmed and validated. To this end, we used an anti-CSNK1G2 monoclonal antibody to pull down the endogenous CSNK1G2 from the testis extracts of 3-month old wild type and CSNK1G2 knockout mice and found that the endogenous RIP3 was only co-precipitated with CSNK1G2 from the extracts of wild type testes. The data confirmed the interaction between the endogenous CSNK1G2 and RIP3 and have been added to the revised manuscript as the new Figure 3D.

2) The role of other CSNK isoforms in RIP3 interactions and necroptosis inhibition is unclear. Mass spectrometry results in Figure 1—figure supplement 1 showed that CSNK1A1, CSNK1D and CSNK1E were pulled down as RIP3-interacting partners. However, CSNK1A did not inhibit RIP3 phosphorylation. Why was that the case? Why was CSNK1G2 not identified in the mass spectrometry experiment? Are any of the other CSNK isoforms expressed in the testis and if so, can they potentially compensate for the loss of CSNK1G2 function in the knockout mice? Since multiple CSNK isoforms could interact with RIP3, do they inhibit RIP3 activation via similar mechanisms in a cell type-specific manner?

CSNK family, with its many members, regulate a variety of cellular signaling pathways. We have thus focused on the function of CSNK1G2, an isoform predominantly expresses in testis. CSNK1D and CSNK1E may also regulate RIP3, a protein is known to be tissue specifically expressed, under different physiological conditions. However, sorting out their regulatory roles is beyond the scope of this paper. The reason for the failure to identify CSNK1G2 in the mass spectrometry experiment in Figure 1—figure supplement 1A is that the experiment was done in HT29 cells, a human colon cancer cell line not expressing CSNK1G2. The observed acceleration of aging in the testis of CSNK1G2 knockout mice does not support the hypothesis that other CSNK isoforms could compensate for the function of CSNK1G2 in this particular tissue. These points are now added to the revised manuscript.

3) The authors show that CSNK deficiency impacts testicular aging and mating performance in older mice. Does CSNK1G2 expression, localization, or activity change with age in wild type mouse or human testes?

We used western blotting and immunohistochemistry to examine the expression, localization, and activity change of CSNK1G2 in wild type mouse testis of different age. As shown in the Author response image 1 (see also Figure 4I), CSNK1G2 expression, localization, and activity do not change with age in wild type mice. The activation of testis necroptosis in aged wild type mouse is not due to the decreased expression/activation of CSNK1G2.

**Author response image 1. sa2fig1:** The expression, localization, and activity change of CSNK1G2 in wild type mouse testis. (A) Western blotting analysis of extracts from phosphate-buffered saline(PBS) perfused testes of *CSNK1G2^+/+^* (WT) mice of 2, 4, 8 and 12-month of age using antibodies against CSNK1G2, RIP3, and GAPDH as indicated.Each group was from a pool of three mice. (B) The expression of RIP3 and CSNK1G2 in mouse testis. The testis sections of 2, 4, 8 and 12-month old wild type mice (n=3) were stained sequentially with antibodies against RIP3 and CSNK1G2 followed by fluorescent-conjugated secondary antibody. Counterstaining with DAPI, blue. Scale bar on the upper panel is 100 μm.

4) Additional detailed fertility data in Figures 4J and 5G would make the author's argument more convincing. Mice at 12-month of age typically show drastically reduced breeding performance, so it is surprising that 12 of out 12 old wild type male mice gave rise to offspring. Was the breeding performance scored over a defined period of time (e.g. within a month)? Another indication for breeding performance is the size of the litter. If the authors have this data from the original breeding experiment, they should be included in the table in Figures 4J and 5G.

The breeding success of 12-month-old male mice was due to the fact that they were mated with 10-week old young female mice. As shown in Figure 4J, 10 out 12 wild type male mice gave rise to offsprings. In Figure 5G, 11 out 12 old wild type male mice gave rise to offsprings, and there was no difference in term of the size of the liter between wild type and CSNK1G knockout mice. Indeed, the female C57BL/6NCrl mouse strain used in this study show drastic lost the breeding performance when reach 12 months old, and the male mice usually lost the reproductive ability at 18-month of age as reported in our previously study (Li et al., 2017).

5) In all figures showing testis histology (3B, 4F, 5D), the panels are too small to evaluate the histology. Higher resolution and larger images are necessary to clearly identify the cell types expressing RIP3 and CSNK1G2 and interpret the testis phenotype.

We used higher resolution and larger imager in the revised figures (Figure 3B, 4F and 5D).

6) At several places in the manuscript, please revise the text to clarify or substantiate conclusions:– In Figure 3B (subsection “Knocking out CSNK1G2 in Testis Cells Significantly Enhanced Their Necroptosis Response”), the authors report expression of CSNK1G2 and RIP3 in spermatogonial stem cells and Sertoli cells, but the (low-resolution) image appears to show expression in multiple spermatogenic cells. Spermatogonial stem cells cannot be identified without specific co-staining; a hematoxylin+IHC serial section image alongside the IF images would be necessary to identify the cell types, and would make the figure easier to interpret. Figure 3C also claims an effect in spermatogonial stem cells and Sertoli cells, but does not provide support for which cell types are being tested. GC2 cells used in Figure 3D-F are derived from spermatocytes (meiotic germ cells), not spermatogonia (pre-meiotic germ cells). These issues could be addressed without affecting the authors' main message by changing "spermatogonium cells" to "spermatogenic cells" throughout the text, to encompass all stages of spermatogenesis.

We agree with the reviewers’ comments and changing "spermatogonium cells" to "spermatogenic cells" in our revised manuscript.

– In Figure 4C, the CSNK1G2 KO mice shown in blue have decreased weight compared to WT. Based on the description in the text and images in Figure 4B, the colors might be mistakenly switched between CSNK1G2 ^-/-^ and ^+/+^. In Figure 4E, the WT and KO labels appear to be swapped.

We are grateful that those mistakes are pointed out by this reviewer and we corrected them in the revised Figure 4C and E.

– The authors should provide numbers and p-values for Figure 1G, and numbers (p-values already provided) for Figures 4A, C, E, H, 5A-C, F, 6C, E, Figure 5—figure supplement 1C, and Figure 5—figure supplement 2A-H.

We provide numbers in the figure legends.

– In Figure 1A, with WT CSNK1G2, Flag-RIP3 shows up as one band, while with the mutant forms, Flag-RIP3 shows up as several bands. Are those high-molecular weight bands the phosphorylated forms, or are there other PTMs of RIP3 that are also affected by CSNK1G2?

RIP3 auto-phosphorylates its S227 to activate necroptosis. Additionally, RIP3 are phosphorylated at multiple sites when ectopically expressed in 293T cells resulting in higher molecular bands shift on SDS-PAGE. The wild type CSNK1G2 was able to inhibit all RIP3 phosphorylation as shown in Figure 1A.

– MEFs express endogenous CSNK1G2 (Figure 1C). Reconstitution of CSNK1G2 in the KO MEFs inhibited RIP3 phosphorylation (Figure 1C). The level of CSNK1G2 expression under this condition was similar to endogenous CSNK1G2 expression in wild type MEFs (Figure 1C, compare lanes 5 and 7). Yet, over-expression in KO MEFs had a stronger effect on RIP3 phosphorylation. Do the authors have an explanation for this?

We also noticed this phenomenon. Our explanation is that the binding and inhibiting of RIP3 by CSNK1G2 is triggered by auto-phosphorylation at serine 211/threonine 215 sites in its C-terminal domain. In MEF cells, we can detect several shift bands of CSNK1G2 using an CSNK1G2 antibody and those bands may be caused by CSNK1G2 auto-phosphorylation. In the recuse cells, a Myc tag was added to the C-terminal of CSNK1G2 and this may promote its auto-phosphorylation, resulting in stronger inhibitory effect of RIP3.

– Throughout the manuscript, a single time point was chosen to demonstrate the effect of CSNK1G2 on RIP3 activation. The effect was not "all or none", suggesting that CSNK1G2 might affect the kinetics of necrosome formation and necroptosis. It would be helpful to discuss how CSNK1G2 might affect the kinetics of the response; this could provide key insights into the mechanism of CSNK1G2 on necroptosis regulation.

We discuss this part in the revised manuscript.

– CSNK1G2 binding to RIP3 seemed to decrease upon TSZ stimulation (Figure 1D). Was this due to de-phosphorylation of CSNK1G2?

This is a good suggestion and we planned to study this observation in the future.